# On Semi-Analytical Solutions for Linearized Dispersive KdV Equations

**Appanah Rao Appadu *** and **Abey Sherif Kelil**

Department of Mathematics and Applied Mathematics, Nelson Mandela University,
Port Elizabeth 6031, South Africa; abeysh2001@gmail.com

**\*** Correspondence: rao.appadu@mandela.ac.za

**Abstract:** The most well-known equations both in the theory of nonlinearity and dispersion, KdV equations, have received tremendous attention over the years and have been used as model equations for the advancement of the theory of solitons. In this paper, some semi-analytic methods are applied to solve linearized dispersive KdV equations with homogeneous and inhomogeneous source terms. These methods are the Laplace-Adomian decomposition method (LADM), Homotopy perturbation method (HPM), Bernstein-Laplace-Adomian Method (BALDM), and Reduced Differential Transform Method (RDTM). Three numerical experiments are considered. As the main contribution, we proposed a new scheme, known as BALDM, which involves Bernstein polynomials, Laplace transform and Adomian decomposition method to solve inhomogeneous linearized dispersive KdV equations. Besides, some modifications of HPM are also considered to solve certain inhomogeneous KdV equations by first constructing a newly modified homotopy on the source term and secondly by modifying Laplace's transform with HPM to build HPTM. Both modifications of HPM numerically confirm the efficiency and validity of the methods for some test problems of dispersive KdV-like equations. We also applied LADM and RDTM to both homogeneous as well as inhomogeneous KdV equations to compare the obtained results and extended to higher dimensions. As a result, RDTM is applied to a 3D-dispersive KdV equation. The proposed iterative schemes determined the approximate solution without any discretization, linearization, or restrictive assumptions. The performance of the four methods is gauged over short and long propagation times and we compute absolute and relative errors at a given time for some spatial nodes.

**Keywords:** Adomian (Laplace) decomposition method; homotopy perturbation method; reduced differential transform method; Bernstein-Laplace-Adomian method; linearized dispersive KdV equation; absolute and relative errors

## 1. Introduction

The well-known Korteweg-de Vries (KdV) equation is a nonlinear dispersive partial differential equation, which describes mathematical modeling of traveling wave solution, known to be solitary water waves (also called solitons) in a shallow water domain. This equation is given by [1]

$$u_t + \gamma u u_x + \beta u_{xxx} = 0. \tag{1}$$

This equation, sometimes known to be the nonlinear evolution equation, was derived in 1895 by Korteweg and de Vries, by the two individuals whose names it bears, in an investigation on shallow water waves, where an exact "solitary wave" solution was also explored [1,2]. The equation was a key achievement in a major controversy on the nature of waves, following the acclaimed "real life"

observation of a solitary wave by John Scott Russell in 1834. For the entire history of the soliton in various existing monographs on solitons and integrable frame works; we refer to the works from Ablowitz and Segur [2], Calogero and Degasperis [3], Ablowitz and Clarkson [4], Drazin [5] and Hirota [6]. This just to know that the KdV equation was revived 70 years after Korteweg and de Vries in a seminal paper by Zabusky and Kruskal [7]. This led to results that were later depicted as a significant development of the twentieth century.

The KdV equation has been an essential tool to describe mathematical modeling and explaining certain events in nature; for example, long internal waves in a ocean [8], acoustic waves on a crystal lattice and magneto-hydrodynamic waves in warm plasma, and ion-acoustic waves in a plasma [9]. In a seminal work by C. Gardner et al. [10], it was shown that the nonlinear PDE given by Equation (1) can be solved by a powerful method, which is known to be the inverse scattering transform method. Despite the fact that this technique can only be applied to very special equations, which we allude to as soliton equations or exactly integrable equations, we presently know entire infinite families of such equations to which the method can be applied.

We quote [11–20] for some recent works on KdV equations and for some applications of inhomogeneous evolution problems, related to KdV-type equations, with application in thermoplastic interaction in a half-space by pulsed laser heating, see [21], where the authors used an eigenvalue approach to get an analytical solution of the inhomogeneous problem under consideration.

The integrability of Equation (1) guarantees an infinite invariants. These quantities are constant along the solution of the given partial differential equation and the first three invariants are [22]

$$F_1(u) = \int_{\mathbb{R}} u \, dx,$$

$$F_2(u) = \frac{1}{2} \int_{\mathbb{R}} u^2 \, dx,$$

$$F_3(u) = \int_{\mathbb{R}} \left( \frac{\beta}{2} (u_x)^2 - \frac{\gamma}{6} u^3 \right) \, dx.$$

The most well-known explicit finite difference scheme for Equation (1) was proposed by Zabusky and Kruskal in 1965. Ascher and McLachlan in [23] gave an account of the study of symplectic and multi-symplectic schemes for KdV equation in order to answer the question of whether added structure preservation such as conservative discretization schemes would provide high quality schemes for long time integration of nonlinear conservative partial differential equations.

Wang et al. [24] proposed a scheme obtained by substituting an average of forward and backward difference in time in place of central difference in time in Zabusky-Kruskal scheme. They carried out numerical simulations of KdV equation with initial equation $u(x, 0) = \cos(x)$ and it was found that their scheme did not blow up at a longer time when compared to the scheme constructed by Zabusky and Kruskal [7], and the multi-symplectic six-point scheme [23]. Appadu et al. [22] proposed two new explicit finite difference schemes for the numerical solutions of KdV equation and analyzed spectral properties of these schemes against two existing schemes proposed by Zabusky and Kruskal [7], Wang et al. [24]. Approximate analytical solutions for linear as well as nonlinear differential equations (for e.g., [25]) can be found using Adomian decomposition method [26–28], Homotopy perturbation method [29,30] and Reduced Differential Transform Method [31,32].

In this paper, we make use of Laplace-Adomian decomposition method (LADM), Homotopy Perturbation method (HPM), and Reduced differential transform method (RDTM) to solve some numerical experiments described by linearized dispersive KdV equations. The methods are derived and results are displayed in Sections 2–4. In Section 5, we applied RDTM for the 3D linearized dispersive KdV equation. Section 6 introduces Bernstein-Adomian Laplace method (BALDM) and

the numerical experiment for the inhomogeneous KdV equation provide us a reliable result. Section 7 ends with discussion and concluding remarks. To measure the efficiency of the semi-analytic method, we used the absolute and relative errors, given by

$$\text{Absolute Error} = |y - \bar{y}|, \tag{2}$$

and

$$\text{Relative Error} = \left| \frac{y - \bar{y}}{y} \right|, \tag{3}$$

where $y$ is the exact value and $\bar{y}$ is the approximate value. The three numerical experiments are detailed below.

**Numerical Experiment 1.** *Numerical experiment 1 Solve the linearized homogeneous dispersive KdV equation [27]*

$$u_t + 2u_x + u_{xxx} = 0, \tag{4}$$

*with $x \in [0, 2\pi]$, and $t \in [0, 4.0]$. The initial condition is $u(x, 0) = \sin(x)$, and the exact solution of this problem is $u(x, t) = \sin(x - t)$.*

**Numerical Experiment 2.** *Solve the linearized inhomogeneous dispersive KdV equation [27]*

$$u_t + u_{xxx} = -\sin(\pi x)\sin(t) - \pi^3 \cos(\pi x)\cos(t), \tag{5}$$

*with $x \in [0, 1]$ and $t \in [0, 0.1]$. The initial condition is $u(x, 0) = \sin(\pi x)$, and the exact solution of this problem is $u(x, t) = \sin(\pi x)\cos(t)$.*

**Numerical Experiment 3.** *Solve the inhomogeneous linearized KdV equation*

$$u_t + xu_x + u_{xxx} = 3xt^2 + 2x + xt^3, \quad x \in [0, 1.0], \quad t \in [0, 1.0], \tag{6}$$

*with initial condition $u_0(x) = 2x$. Exact solution for this problem is $u(x, t) = 2x + xt^3$.*

## 2. Laplace-Adomian Decomposition Method (LADM)

George Adomian [26] found a method to solve linear as well as nonlinear functional equations, which is known to be Adomian decomposition method (ADM) [33–35]. ADM involves partitioning the equation under investigation into linear and nonlinear portions. This method generates a solution in the form of a series whose terms are determined by a recursive relationship using Adomian polynomials [26,28,34]. Some fundamental works on various aspects of the modification of ADM are given by Wazwaz [27] and for some application of ADM to an initial value problem including KdV equations, see [28,33,35] and for Burger's equation, see [36]. We now consider

$$L_t u(x, t) + Ru(x, t) + Nu(x, t) = g(x, t), \tag{7}$$

with an initial condition $u(x, 0) = h(x)$, where $L_t = \frac{\partial}{\partial t}$, $R$ is a linear operator that includes partial derivatives with respect to $x$, $N$ is a nonlinear operator and $g$ is a non-homogeneous term, which is $u$-independent. LADM consists of applying Laplace transform on both sides of Equation (7), obtaining

$$\hat{\mathcal{L}}\left\{ L_t u(x, t) \right\} = \hat{\mathcal{L}}\left\{ g(x, t) - Ru(x, t) - Nu(x, t) \right\}. \tag{8}$$

By applying the inverse Laplace transform to Equation (8) gives

$$u(x,t) = h(x) - \hat{\mathcal{L}}^{-1}\left[\frac{1}{s}\hat{\mathcal{L}}\left\{Ru(x,t) + Nu(x,t)\right\}\right].$$

(9)

LADM method proposes a series solution $u(x,t)$ given by

$$u(x,t) = \sum_{n=0}^{\infty} u_n(x,t).$$

(10)

The nonlinear term $Nu(x,t)$ is given by

$$Nu(x,t) = \sum_{n=0}^{\infty} A_n(u_0, u_1, \ldots, u_n),$$

(11)

where the sequence $\{A_n\}_{n=0}^{\infty}$ are known to be Adomian polynomials, which are given in [26,34,37].
Using Equations (10) and (11) into Equation (9), we obtain

$$\sum_{n=0}^{\infty} u_n(x,t) = h(x) - \hat{\mathcal{L}}^{-1}\left[\frac{1}{s}\hat{\mathcal{L}}\{R\sum_{n=0}^{\infty} u_n(x,t) + \sum_{n=0}^{\infty} A_n(u_0, u_1, \ldots, u_n)\}\right].$$

(12)

From Equation (12), we deduce the following recurrence formulae

$$\begin{cases} u_0(x,t) = h(x), \\ u_{n+1}(x,t) = -\hat{\mathcal{L}}^{-1}\left[\frac{1}{s}\hat{\mathcal{L}}\left\{Ru_n(x,t) + A_n(u_0, u_1, \ldots, u_n)\right\}\right], & n = 0,1,2,\ldots. \end{cases}$$

(13)

Using Equation (13), an approximate solution of Equation (7) is obtained using

$$u(x,t) \approx \sum_{n=0}^{k} u_n(x,t), \quad \text{where} \quad \lim_{k\to\infty} \sum_{n=0}^{k} u_n(x,t) = u(x,t).$$

(14)

**Remark 1.** *In most cases, the results obtained by LADM and ADM are exactly the same for linear homogeneous PDEs. However, LADM, in some sense is a modification of ADM that needs less work in comparison to ADM as the latter involves evaluation of Adomian polynomials and the former employs Laplace transform [38]. Without linearising the problem, LADM decreases considerably huge volume of calculations.*

### 2.1. Solution of Numerical Experiment 1 via LADM

Consider the linearized KdV equation in Equation (4) in its standard form as

$$\mathsf{L}_t u + 2u_x + u_{xxx} = 0, \quad x \in [0, 2\pi], \quad t \in [0, 4.0],$$

(15)

with the differential operator $\mathsf{L}_t = \dfrac{\partial}{\partial t}$. The initial condition is given by

$$u(x,0) = \sin(x).$$

(16)

By applying the Laplace transform $\hat{\mathcal{L}}$ on Equation (15), we obtain

$$u(x,s) = \frac{u(x,0)}{s} - \frac{2}{s}\hat{\mathcal{L}}\{u_x\} - \frac{1}{s}\hat{\mathcal{L}}\{u_{xxx}\}.$$

(17)

Taking inverse Laplace transform of Equation (17), we get

$$u(x,t) = u(x,0) - \hat{\mathcal{L}}^{-1}\left[\frac{1}{s}[\hat{\mathcal{L}}\{2u_x\} - \hat{\mathcal{L}}\{u_{xxx}\}]\right].$$

(18)

By decomposing $u(x,t) = \sum_{n \geq 0} u_n(x,t)$ in Equation (18), we obtain the recursive relation as

$$
\left.
\begin{aligned}
u_0(x) &= u(x,0) = \sin(x), \\
u_1(x,t) &= -\mathcal{L}^{-1}\left[\frac{1}{s}\left[\mathcal{L}\{-2u_{0,x}\} - \mathcal{L}\{u_{0,xxx}\}\right]\right], \\
u_2(x,t) &= -\mathcal{L}^{-1}\left[\frac{1}{s}\left[\mathcal{L}\{-2u_{1,x}\} - \mathcal{L}\{u_{1,xxx}\}\right]\right], \\
&\vdots \\
u_n(x,t) &= -\mathcal{L}^{-1}\left[\frac{1}{s}\left[\mathcal{L}\{-2u_{n-1,x}\} - \mathcal{L}\{u_{n-1,xxx}\}\right]\right],
\end{aligned}
\right\}
\tag{19}
$$

where $u_{j,y}$ denotes the *j*th-derivative of $u$ with respect to $y$.

By using Equation (19) with the initial condition $u(x,0) = \sin(x)$, together with properties of Laplace's transform [38], the first few LADM components of $u_n(x,t)$ for $n \geq 1$ are given by

$$
\left.
\begin{aligned}
u_1(x,t) &= -t\cos(x), \quad u_2(x,t) = -\frac{t^2}{2!}\sin(x), \quad u_3(x,t) = \frac{t^3}{3!}\cos(x), \\
u_4(x,t) &= \frac{t^4}{4!}\sin(x), \quad u_5(x,t) = -\frac{t^5}{5!}\cos(x), \quad u_6(x,t) = -\frac{t^6}{6!}\sin(x), \\
u_7(x,t) &= \frac{t^7}{7!}\cos(x), \quad u_8(x,t) = \frac{t^8}{8!}\sin(x), \\
u_9(x,t) &= -\frac{t^9}{9!}\cos(x), \qquad u_{10}(x,t) = -\frac{t^{10}}{10!}\sin(x),
\end{aligned}
\right\}
\tag{20}
$$

and so on.

The tenth-term approximate LADM solution is given by

$$
\begin{aligned}
\Psi_{10}(x,t) = \sum_{i=0}^{10} u_i(x,t) &= \left(\sin(x) - \frac{t^2}{2!}\sin(x) + \frac{t^4}{4!}\sin(x) - \frac{t^6}{6!}\sin(x) + \frac{t^8}{8!}\sin(x) - \frac{t^{10}}{10!}\sin(x)\right) \\
&+ \left(-t\cos(x) + \frac{t^3}{3!}\cos(x) - \frac{t^5}{5!}\cos(x) + \frac{t^7}{7!}\cos(x) - \frac{t^9}{9!}\cos(x)\right).
\end{aligned}
\tag{21}
$$

Thus, using the convergence property of LADM [28], we have, for any $n \in \mathbb{N}_0$, that

$$
\begin{aligned}
u(x,t) &= \sin(x)\left[1 - \frac{t^2}{2!} + \frac{t^4}{4!} - \frac{t^6}{6!} + \frac{t^8}{8!} + \ldots\right] - \cos(x)\left[t - \frac{t^3}{3!} + \frac{t^5}{5!} - \frac{t^7}{7!} + \ldots\right], \\
&= \sin(x)\cos(t) - \cos(x)\sin(t) = \sin(x - t),
\end{aligned}
$$

the closed form as required.

**Remark 2.** *We note that LADM, HPM and RDTM are equivalent schemes when applied to linearized homogeneous KdV equation. We also assumed that $t = 4.0$ as a long propagating time for the homogeneous linearized dispersive KdV equation whereas $t = 0.1$ as short propagating time for the inhomogeneous dispersive KdV equation in Section 2.2.*

We obtain plots of exact and approximate solution using LADM, HPM, RDTM vs. $x$ at times 0.1, 2.0 and 4.0 in Figure 1. We also tabulate absolute and relative errors at some values of $x$ at times 0.1, 2.0, 4.0 using LADM, HPM, RDTM in Table 1 and see also Figure 2.

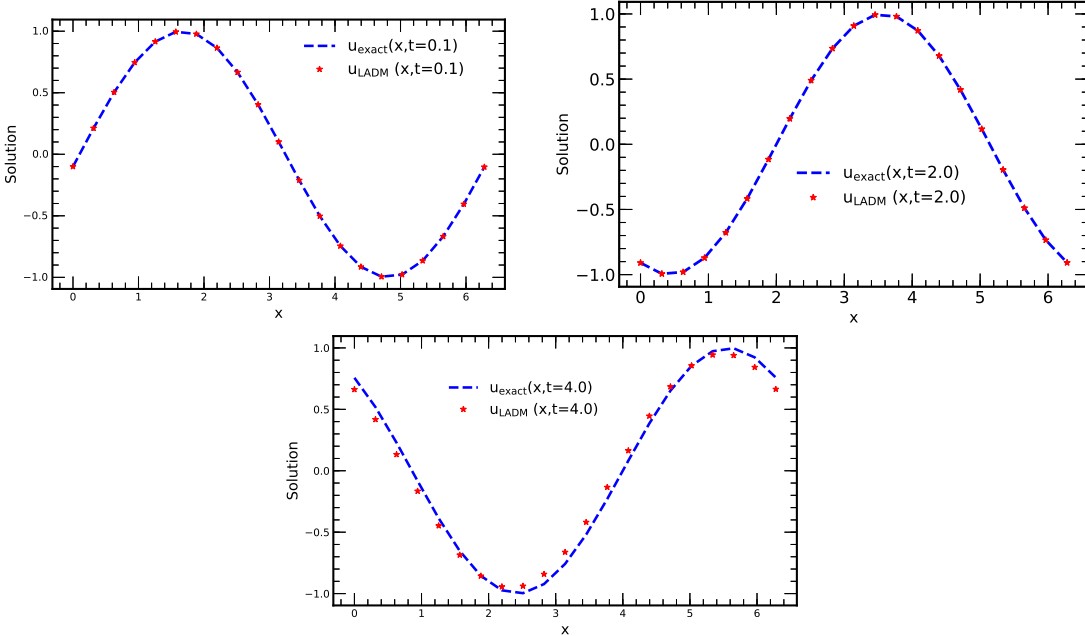

**Figure 1.** Plots of exact solution and approximate solution using 10-terms of LADM, HPM, RDTM vs. *x* at times 0.1, 2.0 and 4.0. (The space interval used for these plots is $\frac{\pi}{10} \approx 0.314$).

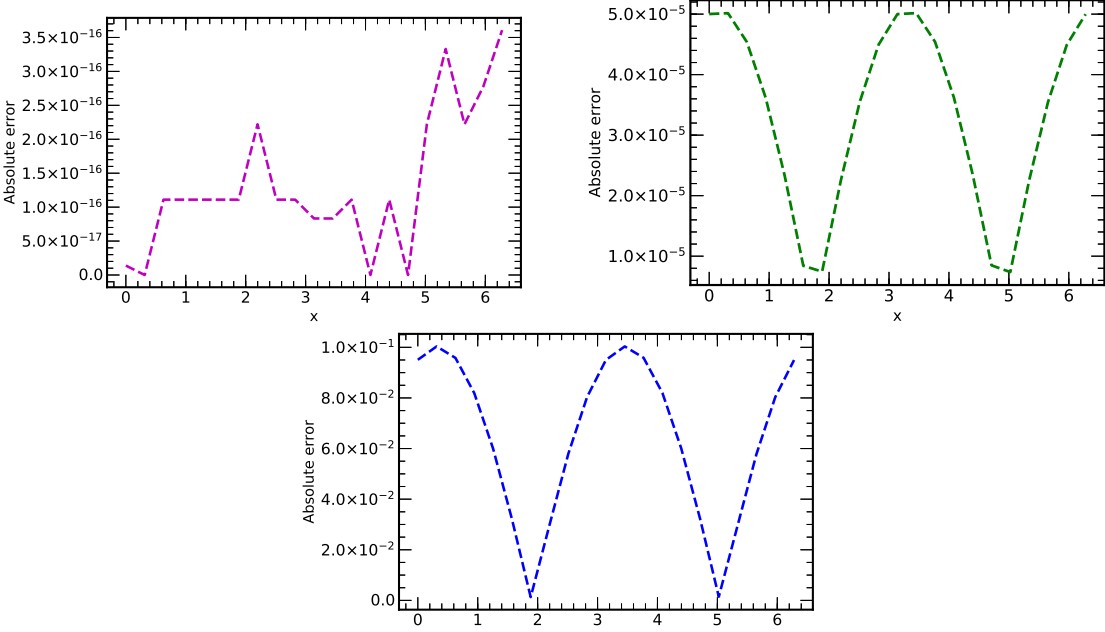

**Figure 2.** Plots of absolute errors vs. *x* at different values of time ($t = 0.1, 2.0, 4.0$) using the methods LADM, HPM and RDTM.

**Table 1.** Absolute and relative errors at some values of $x$ obtained at times 0.1, 2.0, 4.0 using 10-terms of LADM, HPM and RDTM for numerical experiment 1.

| $t$ | Values of $x$ | Exact Solution | Numerical Solution | Absolute Error | Relative Error |
|---|---|---|---|---|---|
| | 0.314 | 0.212370 | 0.212370 | 0.000000 | 0.000000 |
| | 0.628 | 0.503807 | 0.503807 | $1.110223 \times 10^{-16}$ | $2.203669 \times 10^{-16}$ |
| | 0.942 | 0.745977 | 0.745977 | $1.110223 \times 10^{-16}$ | $1.488281 \times 10^{-16}$ |
| | 1.256 | 0.915198 | 0.915198 | $1.110223 \times 10^{-16}$ | $1.213095 \times 10^{-16}$ |
| | 1.570 | 0.994924 | 0.994924 | $1.110223 \times 10^{-16}$ | $1.115887 \times 10^{-16}$ |
| | 1.884 | 0.977358 | 0.977358 | $1.110223 \times 10^{-16}$ | $1.135943 \times 10^{-16}$ |
| | 2.198 | 0.864217 | 0.864217 | $2.220446 \times 10^{-16}$ | $2.569314 \times 10^{-16}$ |
| | 2.512 | 0.666566 | 0.666566 | $1.110223 \times 10^{-16}$ | $1.665586 \times 10^{-16}$ |
| $t = 0.1$ | 2.826 | 0.403732 | 0.403732 | $1.110223 \times 10^{-16}$ | $2.749900 \times 10^{-16}$ |
| | 3.140 | 0.101418 | 0.101418 | $8.326673 \times 10^{-17}$ | $8.210252 \times 10^{-16}$ |
| | 3.454 | $-0.210814$ | $-0.210814$ | $8.326673 \times 10^{-17}$ | $3.949777 \times 10^{-16}$ |
| | 3.768 | $-0.502430$ | $-0.502430$ | $1.110223 \times 10^{-16}$ | $2.209705 \times 10^{-16}$ |
| | 4.082 | $-0.744915$ | $-0.744915$ | 0.000000 | 0.000000 |
| | 4.396 | $-0.914555$ | $-0.914555$ | $1.110223 \times 10^{-16}$ | $1.213948 \times 10^{-16}$ |
| | 4.710 | $-0.994763$ | $-0.994763$ | 0.000000 | 0.000000 |
| | 5.024 | $-0.977694$ | $-0.977694$ | $2.220446 \times 10^{-16}$ | $2.271106 \times 10^{-16}$ |
| | 5.338 | $-0.865018$ | $-0.865018$ | $3.330669 \times 10^{-16}$ | $3.850406 \times 10^{-16}$ |
| | 5.652 | $-0.667752$ | $-0.667752$ | $2.220446 \times 10^{-16}$ | $3.325253 \times 10^{-16}$ |
| | 5.966 | $-0.405189$ | $-0.405189$ | $2.775558 \times 10^{-16}$ | $6.850036 \times 10^{-16}$ |
| | 6.280 | $-0.103002$ | $-0.103002$ | $3.608225 \times 10^{-16}$ | $3.503053 \times 10^{-15}$ |
| | 0.314 | $-0.993371$ | $-0.993422$ | $5.015442 \times 10^{-5}$ | $5.048909 \times 10^{-5}$ |
| | 0.628 | $-0.980305$ | $-0.980350$ | $4.538846 \times 10^{-5}$ | $4.630034 \times 10^{-5}$ |
| | 0.942 | $-0.871376$ | $-0.871412$ | $3.618402 \times 10^{-5}$ | $4.152515 \times 10^{-5}$ |
| | 1.256 | $-0.677236$ | $-0.677260$ | $2.344120 \times 10^{-5}$ | $3.461303 \times 10^{-5}$ |
| | 1.570 | $-0.416871$ | $-0.416879$ | $8.406101 \times 10^{-6}$ | $2.016476 \times 10^{-5}$ |
| | 1.884 | $-0.115740$ | $-0.115733$ | $7.451020 \times 10^{-6}$ | $6.437721 \times 10^{-5}$ |
| | 2.198 | 0.196709 | 0.196731 | $2.257952 \times 10^{-5}$ | $1.147865 \times 10^{-4}$ |
| | 2.512 | 0.489922 | 0.489957 | $3.549999 \times 10^{-5}$ | $7.246054 \times 10^{-5}$ |
| $t = 2.0$ | 2.826 | 0.735226 | 0.735271 | $4.494898 \times 10^{-5}$ | $6.113628 \times 10^{-5}$ |
| | 3.140 | 0.908633 | 0.908683 | $5.000247 \times 10^{-5}$ | $5.503040 \times 10^{-5}$ |
| | 3.454 | 0.993187 | 0.993237 | $5.016629 \times 10^{-5}$ | $5.051041 \times 10^{-5}$ |
| | 3.768 | 0.980618 | 0.980664 | $4.542442 \times 10^{-5}$ | $4.632222 \times 10^{-5}$ |
| | 4.082 | 0.872156 | 0.872193 | $3.624056 \times 10^{-5}$ | $4.155283 \times 10^{-5}$ |
| | 4.396 | 0.678407 | 0.678431 | $2.351279 \times 10^{-5}$ | $3.465881 \times 10^{-5}$ |
| | 4.710 | 0.418318 | 0.418326 | $8.485738 \times 10^{-6}$ | $2.028538 \times 10^{-5}$ |
| | 5.024 | 0.117322 | 0.117314 | $7.371122 \times 10^{-6}$ | $6.282822 \times 10^{-5}$ |
| | 5.338 | $-0.195147$ | $-0.195170$ | $2.250717 \times 10^{-5}$ | $1.153344 \times 10^{-4}$ |
| | 5.652 | $-0.488533$ | $-0.488568$ | $3.544228 \times 10^{-5}$ | $7.254842 \times 10^{-5}$ |
| | 5.966 | $-0.734146$ | $-0.734190$ | $4.491153 \times 10^{-5}$ | $6.117524 \times 10^{-5}$ |
| | 6.280 | $-0.907967$ | $-0.908017$ | $4.998895 \times 10^{-5}$ | $5.505590 \times 10^{-5}$ |
| | 0.314 | 0.517911 | 0.417558 | $1.003528 \times 10^{-1}$ | $1.937646 \times 10^{-1}$ |
| | 0.628 | 0.228374 | 0.132556 | $9.581821 \times 10^{-2}$ | $4.195668 \times 10^{-1}$ |
| | 0.942 | $-0.083495$ | $-0.165409$ | $8.191365 \times 10^{-2}$ | $9.810567 \times 10^{-1}$ |
| | 1.256 | $-0.387200$ | $-0.447199$ | $5.999889 \times 10^{-2}$ | $1.549558 \times 10^{-1}$ |
| | 1.570 | $-0.653041$ | $-0.685258$ | $3.221691 \times 10^{-2}$ | $4.933369 \times 10^{-2}$ |
| | 1.884 | $-0.855022$ | $-0.856306$ | $1.284492 \times 10^{-3}$ | $1.502292 \times 10^{-3}$ |
| | 2.198 | $-0.973391$ | $-0.943618$ | $2.977354 \times 10^{-2}$ | $3.058743 \times 10^{-2}$ |
| | 2.512 | $-0.996574$ | $-0.938654$ | $5.792005 \times 10^{-2}$ | $5.811915 \times 10^{-2}$ |
| | 2.826 | $-0.922304$ | $-0.841901$ | $8.040265 \times 10^{-2}$ | $8.717588 \times 10^{-2}$ |
| $t = 4.0$ | 3.140 | $-0.757843$ | $-0.662820$ | $9.502279 \times 10^{-2}$ | $1.253859 \times 10^{-1}$ |
| | 3.454 | $-0.519273$ | $-0.418922$ | $1.003508 \times 10^{-1}$ | $1.932525 \times 10^{-1}$ |
| | 3.768 | $-0.229924$ | $-0.134059$ | $9.586562 \times 10^{-2}$ | $4.169441 \times 10^{-1}$ |
| | 4.082 | 0.081908 | 0.163914 | $8.200590 \times 10^{-2}$ | $0.1001194 \times 10^{1}$ |
| | 4.396 | 0.385731 | 0.445858 | $6.012694 \times 10^{-2}$ | $1.558779 \times 10^{-1}$ |
| | 4.710 | 0.651834 | 0.684202 | $3.236825 \times 10^{-2}$ | $4.965722 \times 10^{-2}$ |
| | 5.024 | 0.854195 | 0.855639 | $1.444316 \times 10^{-3}$ | $1.690851 \times 10^{-3}$ |
| | 5.338 | 0.973025 | 0.943404 | $2.962086 \times 10^{-2}$ | $3.044203 \times 10^{-2}$ |
| | 5.652 | 0.996705 | 0.938915 | $5.778945 \times 10^{-2}$ | $5.798050 \times 10^{-2}$ |
| | 5.966 | 0.922918 | 0.842611 | $8.030689 \times 10^{-2}$ | $8.701410 \times 10^{-2}$ |
| | 6.280 | 0.758881 | 0.663909 | $9.497124 \times 10^{-2}$ | $1.251465 \times 10^{-1}$ |

**Remark 3.** *Table 1 shows that the approximate solutions of Equation (4) deviate from the exact solution at a larger value of t as also shown in Figure 3; for example, at t = 0.628, the absolute errors are in orders of $10^{-16}$; at t = 2.0, the absolute errors are in orders of $10^{-6}$ to $10^{-5}$, and at t = 4.0, the absolute errors are in orders of $10^{-2}$ to $10^{-1}$. (see also Figures 1 and 2 for the graphical demonstration).*

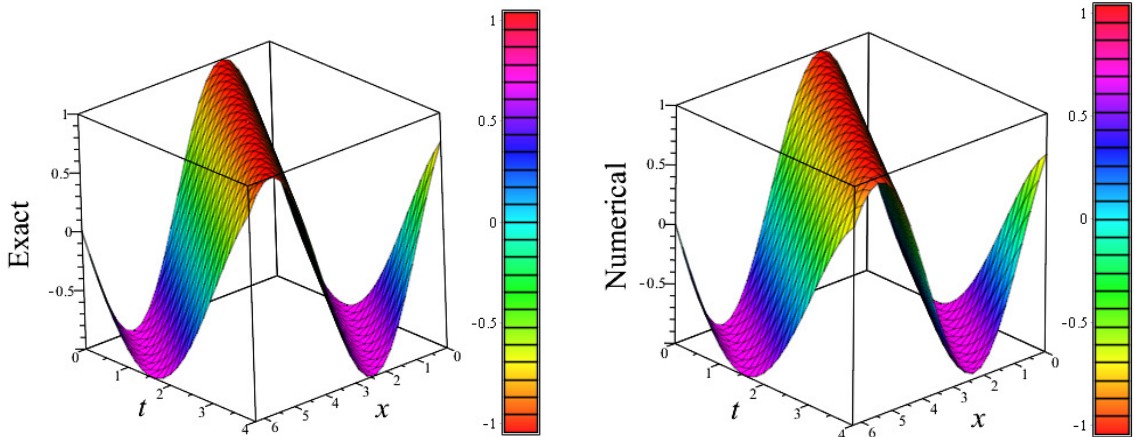

**Figure 3.** 3D plots of exact solution and approximate solution (using LADM, HPM, RDTM), vs. *x* vs. *t*, for numerical experiment 1.

### 2.2. Solution of Numerical Experiment 2 via LADM

The one-dimensional inhomogeneous dispersive PDE [27] is given by

$$u_t + u_{xxx} = g(x,t). \tag{22}$$

We consider $x \in [0, 0.1]$, $t > 0$ with source term $g(x,t) = -\sin(\pi x)\sin(t) - \pi^3 \cos(\pi x)\cos(t)$ and the initial condition is $u(x,0) = \sin(\pi x)$. The time dependent boundary conditions are $u(0,t) = 0$ and $u_x(0,t) = \pi \cos(t)$. Exact solution for this problem is given by

$$u(x,t) = \sin(\pi x)\cos(t). \tag{23}$$

By applying Equations (8)–(10) and employing the initial condition $u(x,0) = \sin(\pi x)$ together with Equation (10), we obtain

$$\sum_{n\geq 0} u_n(x,t) = \hat{\mathcal{L}}^{-1}\left(\frac{u(x,0)}{s}\right) + \hat{\mathcal{L}}^{-1}\left[\frac{1}{s}\left\{-\hat{\mathcal{L}}\left[\sum_{n\geq 0} u_{n,xxx}^3(x,t)\right] + \hat{\mathcal{L}}\{-\sin(\pi x)\,\sin(t) - \pi^3 \cos(\pi x)\,\cos(t)\}\right\}\right]. \tag{24}$$

The following recursive relation is derived from the components of the series solution in Equation (24):

$$\left.\begin{array}{l} u_0(x,t) = u(x,0) + \hat{\mathcal{L}}^{-1}\left[\frac{1}{s}\mathcal{L}\left[g(x,t)\right]\right], \\[2ex] u_1(x,t) = -\hat{\mathcal{L}}^{-1}\left[\frac{1}{s}\hat{\mathcal{L}}\left[\frac{\partial u_0^3(x,t)}{\partial x^3}\right]\right], \\[2ex] \vdots \\[2ex] u_n(x,t) = -\hat{\mathcal{L}}^{-1}\left[\frac{1}{s}\mathcal{L}\left[\frac{\partial u_{n-1}^3(x,t)}{\partial x^3}\right]\right], \quad n \geq 1. \end{array}\right\} \tag{25}$$

Thus, the first few LADM solutions, using Equation (25), are given by

$$u_0(x,t) = u(x,0) + \hat{\mathcal{L}}^{-1}\left[\frac{1}{s}\mathcal{L}\left[g(x,t)\right]\right] = -\pi^3 \cos(\pi x)\sin(t) + \sin(\pi x)\cos(t), \tag{26a}$$

$$u_1(x,t) = -\hat{\mathcal{L}}^{-1}\left[\frac{1}{s}\hat{\mathcal{L}}\left[\frac{\partial u_0^3(x,t)}{\partial x^3}\right]\right] = \pi^3 \cos(\pi x)\sin(t) - \pi^6 \sin(\pi x)\cos(t) + \pi^6 \sin(\pi x), \tag{26b}$$

$$u_2(x,t) = -\hat{\mathcal{L}}^{-1}\left[\frac{1}{s}\hat{\mathcal{L}}\left[\frac{\partial u_1^3(x,t)}{\partial x^3}\right]\right] = \pi^9 \cos(\pi x)\left[t - \sin(t)\right] + \pi^6 \sin(\pi x)\left[\cos(t) - 1\right], \tag{26c}$$

$$u_3(x,t) = -\hat{\mathcal{L}}^{-1}\left[\frac{1}{s}\hat{\mathcal{L}}\left[\frac{\partial u_2^3(x,t)}{\partial x^3}\right]\right] = \pi^9 \cos(\pi x)\left[\sin(t) - t\right] + \pi^{12}\sin(\pi x)\left[1 - \cos(t) - \frac{t^2}{2!}\right], \tag{26d}$$

$$u_4(x,t) = -\hat{\mathcal{L}}^{-1}\left[\frac{1}{s}\hat{\mathcal{L}}\left[\frac{\partial u_3^3(x,t)}{\partial x^3}\right]\right] = \pi^{15}\cos(\pi x)\left[t - \sin(t) - \frac{t^3}{3!}\right] - \pi^{12}\sin(\pi x)\left[1 - \cos(t) - \frac{t^2}{2!}\right], \tag{26e}$$

$$u_5(x,t) = -\hat{\mathcal{L}}^{-1}\left[\frac{1}{s}\hat{\mathcal{L}}\left[\frac{\partial u_4^3(x,t)}{\partial x^3}\right]\right] = \pi^{15}\cos(\pi x)\left[\frac{t^3}{3!} + \sin(t) - t\right] + \pi^{18}\sin(\pi x)\left[1 - \cos(t) - \frac{t^2}{2!} + \frac{t^4}{4!}\right], \tag{26f}$$

$$u_6(x,t) = -\hat{\mathcal{L}}^{-1}\left[\frac{1}{s}\hat{\mathcal{L}}\left[\frac{\partial u_5^3(x,t)}{\partial x^3}\right]\right] = \pi^{21}\cos(\pi x)\left[t - \sin(t) - \frac{t^3}{3!} + \frac{t^5}{5!}\right] - \pi^{18}\sin(\pi x)\left[1 - \cos(t) - \frac{t^2}{2!} + \frac{t^4}{4!}\right], \tag{26g}$$

$$u_7(x,t) = -\hat{\mathcal{L}}^{-1}\left[\frac{1}{s}\hat{\mathcal{L}}\left[\frac{\partial u_6^3(x,t)}{\partial x^3}\right]\right] = -\pi^{21}\cos(\pi x)\left[t - \frac{t^3}{3!} + \frac{t^5}{5!} - \sin(t)\right] + \pi^{24}\sin(\pi x)\left[1 - \frac{t^2}{2!} - \cos(t) - \frac{t^4}{4!} - \frac{t^6}{6!}\right]. \tag{26h}$$

From the fact in Equation (14), the series of approximate solutions, Equation (26a–h), takes the form

$$\Psi_7(x,t) = \sin(\pi x)\cos(t) - \pi^{24}\sin(\pi x)\cos(t) + \pi^{24}\left(1 - \frac{t^2}{2!} + \frac{t^4}{4!} - \frac{t^6}{6!}\right)\sin(\pi x) + \dots, \tag{27}$$

which converges to the exact solution in Equation (23).

Table 2 demonstrates absolute and relative errors using LADM for the non-homogeneous dispersive Linearized KdV equation, showing how the approximate solution is compared with the exact solution. See Figures 4 and 5 for pictorial representation of the solution as well as the errors and Figure 6 for 3D plot of the exact solution as well as LADM solution.

**Remark 4.** *One can observe from Equation (26a–h) the occurrence of noise terms. By 'noise' terms, we mean the identical terms, with opposite signs, which may appear in various components $u_j$, $j \geq 1$ [37,39]. G. Adomian and Rach in [40] and Wazwaz [37] investigated the phenomenon of self-cancelling 'noise' terms where some terms in the series vanish on the limit. These 'noise' terms do not show up for homogeneous equations but solely for specific types of inhomogeneous equations. It was formally shown that by canceling the noise terms that appear in $u_0$ and $u_1$ from $u_0$, even though $u_1$ contains additional terms, the remaining non-cancelled terms of $u_0$ may give the exact solution of the inhomogeneous problem [35,41]. This can be justified through substitution. Therefore, it is necessary to verify that the non-cancelled terms of $u_0$ satisfy the PDE under discussion, which holds in our case. A necessary condition for the generation of the noise terms for inhomogeneous problems is that the zeroth component $u_0$ must contain the exact solution $u$ among other terms. For a complete and thorough study on noise terms, please refer to [39].*

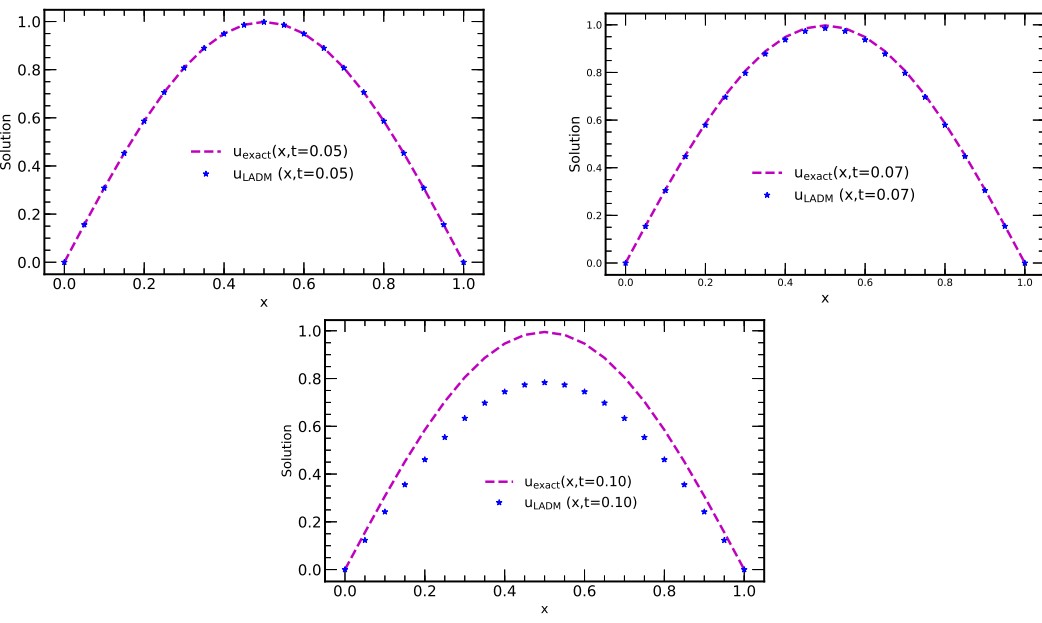

**Figure 4.** Plots of exact solution and solution using LADM vs. $x$ at different $t$-values, $t = 0.05, 0.07, 0.10$ (for numerical experiment 2).

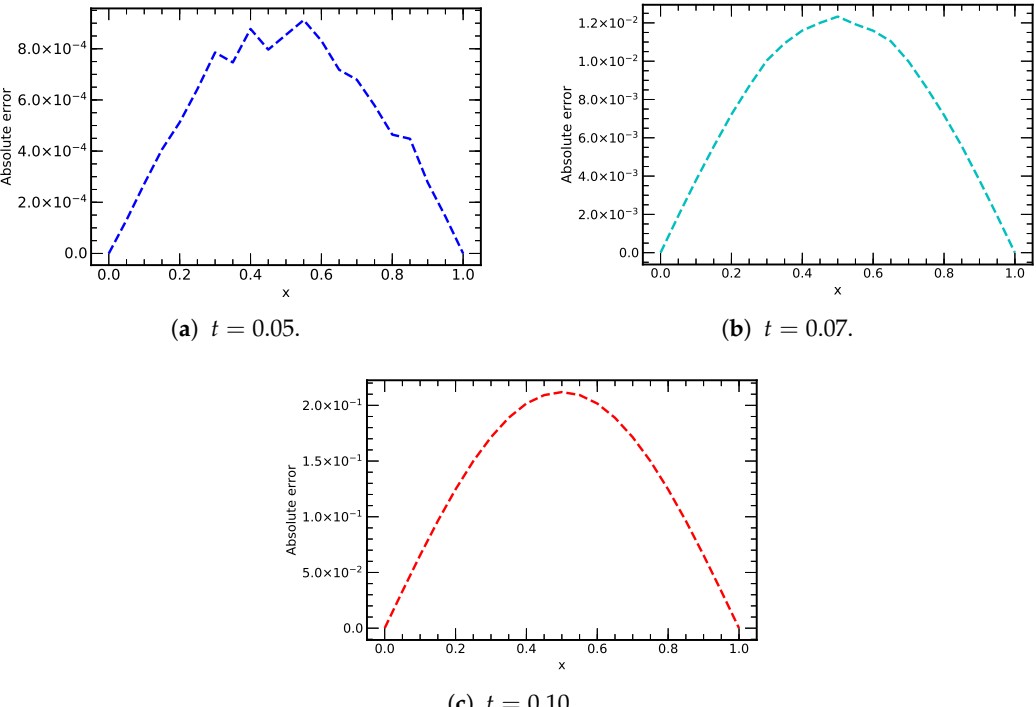

**Figure 5.** Plots of absolute errors vs. $x$ using 7-terms of LADM at different values of time ($t = 0.05, 0.07, 0.10$).

**Table 2.** Absolute and relative errors at some values of $x$ obtained at times 0.05, 0.07, 0.10 using 7-terms of LADM (for numerical experiment 2).

| $t$ | Values of $x$ | Exact Solution | Numerical Solution | Absolute Error | Relative Error |
|---|---|---|---|---|---|
| | 0.050 | 0.156239 | 0.156108 | $1.306562 \times 10^{-4}$ | $8.362586 \times 10^{-4}$ |
| | 0.100 | 0.308631 | 0.308359 | $2.718451 \times 10^{-4}$ | $8.808101 \times 10^{-4}$ |
| | 0.150 | 0.453423 | 0.453017 | $4.062699 \times 10^{-4}$ | $8.960062 \times 10^{-4}$ |
| | 0.200 | 0.587051 | 0.586538 | $5.123988 \times 10^{-4}$ | $8.728357 \times 10^{-4}$ |
| | 0.250 | 0.706223 | 0.705581 | $6.422466 \times 10^{-4}$ | $9.094104 \times 10^{-4}$ |
| | 0.300 | 0.808006 | 0.807220 | $7.856065 \times 10^{-4}$ | $9.722781 \times 10^{-4}$ |
| | 0.350 | 0.889893 | 0.889146 | $7.466506 \times 10^{-4}$ | $8.390341 \times 10^{-4}$ |
| | 0.400 | 0.949868 | 0.948990 | $8.776868 \times 10^{-4}$ | $9.240093 \times 10^{-4}$ |
| | 0.450 | 0.986454 | 0.985657 | $7.967306 \times 10^{-4}$ | $8.076713 \times 10^{-4}$ |
| $t = 0.05$ | 0.500 | 0.998750 | 0.997896 | $8.544922 \times 10^{-4}$ | $8.555614 \times 10^{-4}$ |
| | 0.550 | 0.986454 | 0.985542 | $9.121347 \times 10^{-4}$ | $9.246602 \times 10^{-4}$ |
| | 0.600 | 0.949868 | 0.949037 | $8.314168 \times 10^{-4}$ | $8.752972 \times 10^{-4}$ |
| | 0.650 | 0.889893 | 0.889175 | $7.179544 \times 10^{-4}$ | $8.067874 \times 10^{-4}$ |
| | 0.700 | 0.808006 | 0.807327 | $6.791179 \times 10^{-4}$ | $8.404863 \times 10^{-4}$ |
| | 0.750 | 0.706223 | 0.705645 | $5.784568 \times 10^{-4}$ | $8.190851 \times 10^{-4}$ |
| | 0.800 | 0.587051 | 0.586586 | $4.644615 \times 10^{-4}$ | $7.911779 \times 10^{-4}$ |
| | 0.850 | 0.453423 | 0.452975 | $4.482223 \times 10^{-4}$ | $9.885298 \times 10^{-4}$ |
| | 0.900 | 0.308631 | 0.308353 | $2.775309 \times 10^{-4}$ | $8.992326 \times 10^{-4}$ |
| | 0.950 | 0.156239 | 0.156095 | $1.442404 \times 10^{-4}$ | $9.232041 \times 10^{-4}$ |
| | 1.000 | 0.000000 | 0.000000 | $1.386416 \times 10^{-17}$ | $1.133511 \times 10^{-1}$ |
| | 0.050 | 0.156051 | 0.154127 | $1.924276 \times 10^{-3}$ | $1.233105 \times 10^{-2}$ |
| | 0.100 | 0.308260 | 0.304473 | $3.786796 \times 10^{-3}$ | $1.228441 \times 10^{-2}$ |
| | 0.150 | 0.452879 | 0.447335 | $5.543939 \times 10^{-3}$ | $1.224155 \times 10^{-2}$ |
| | 0.200 | 0.586346 | 0.579125 | $7.220799 \times 10^{-3}$ | $1.231492 \times 10^{-2}$ |
| | 0.250 | 0.705375 | 0.696679 | $8.695814 \times 10^{-3}$ | $1.232793 \times 10^{-2}$ |
| | 0.300 | 0.807036 | 0.796997 | $1.003879 \times 10^{-2}$ | $1.243910 \times 10^{-2}$ |
| | 0.350 | 0.888824 | 0.877889 | $1.093497 \times 10^{-2}$ | $1.230273 \times 10^{-2}$ |
| | 0.400 | 0.948727 | 0.937126 | $1.160181 \times 10^{-2}$ | $1.222881 \times 10^{-2}$ |
| | 0.450 | 0.985269 | 0.973269 | $1.200080 \times 10^{-2}$ | $1.218022 \times 10^{-2}$ |
| $t = 0.07$ | 0.500 | 0.997551 | 0.985222 | $1.232910 \times 10^{-2}$ | $1.235937 \times 10^{-2}$ |
| | 0.550 | 0.985269 | 0.973345 | $1.192498 \times 10^{-2}$ | $1.210326 \times 10^{-2}$ |
| | 0.600 | 0.948727 | 0.937136 | $1.159155 \times 10^{-2}$ | $1.221800 \times 10^{-2}$ |
| | 0.650 | 0.888824 | 0.877787 | $1.103793 \times 10^{-2}$ | $1.241857 \times 10^{-2}$ |
| | 0.700 | 0.807036 | 0.797055 | $9.980618 \times 10^{-3}$ | $1.236701 \times 10^{-2}$ |
| | 0.750 | 0.705375 | 0.696737 | $8.638171 \times 10^{-3}$ | $1.224621 \times 10^{-2}$ |
| | 0.800 | 0.586346 | 0.579162 | $7.183557 \times 10^{-3}$ | $1.225140 \times 10^{-2}$ |
| | 0.850 | 0.452879 | 0.447314 | $5.564460 \times 10^{-3}$ | $1.228687 \times 10^{-2}$ |
| | 0.900 | 0.308260 | 0.304479 | $3.781563 \times 10^{-3}$ | $1.226744 \times 10^{-2}$ |
| | 0.950 | 0.156051 | 0.154130 | $1.920938 \times 10^{-3}$ | $1.230966 \times 10^{-2}$ |
| | 1.000 | 0.000000 | 0.000000 | $2.502025 \times 10^{-17}$ | $2.048074 \times 10^{-1}$ |
| | 0.050 | 0.155653 | 0.122508 | $3.314447 \times 10^{-2}$ | $2.129383 \times 10^{-1}$ |
| | 0.100 | 0.307473 | 0.242008 | $6.546499 \times 10^{-2}$ | $2.129128 \times 10^{-1}$ |
| | 0.150 | 0.451722 | 0.355480 | $9.624288 \times 10^{-2}$ | $2.130576 \times 10^{-1}$ |
| | 0.200 | 0.584849 | 0.460271 | $1.245775 \times 10^{-1}$ | $2.130081 \times 10^{-1}$ |
| | 0.250 | 0.703574 | 0.553690 | $1.498841 \times 10^{-1}$ | $2.130323 \times 10^{-1}$ |
| | 0.300 | 0.804975 | 0.633573 | $1.714020 \times 10^{-1}$ | $2.129283 \times 10^{-1}$ |
| | 0.350 | 0.886555 | 0.697687 | $1.888679 \times 10^{-1}$ | $2.130357 \times 10^{-1}$ |
| | 0.400 | 0.946305 | 0.744603 | $2.017019 \times 10^{-1}$ | $2.131467 \times 10^{-1}$ |
| | 0.450 | 0.982754 | 0.773581 | $2.091732 \times 10^{-1}$ | $2.128439 \times 10^{-1}$ |
| | 0.500 | 0.995004 | 0.783090 | $2.119141 \times 10^{-1}$ | $2.129781 \times 10^{-1}$ |
| $t = 0.10$ | 0.550 | 0.982754 | 0.773470 | $2.092838 \times 10^{-1}$ | $2.129565 \times 10^{-1}$ |
| | 0.600 | 0.946305 | 0.744687 | $2.016184 \times 10^{-1}$ | $2.130586 \times 10^{-1}$ |
| | 0.650 | 0.886555 | 0.697738 | $1.888175 \times 10^{-1}$ | $2.129789 \times 10^{-1}$ |
| | 0.700 | 0.804975 | 0.633604 | $1.713715 \times 10^{-1}$ | $2.128904 \times 10^{-1}$ |
| | 0.750 | 0.703574 | 0.553653 | $1.499208 \times 10^{-1}$ | $2.130845 \times 10^{-1}$ |
| | 0.800 | 0.584849 | 0.460341 | $1.245074 \times 10^{-1}$ | $2.128883 \times 10^{-1}$ |
| | 0.850 | 0.451722 | 0.355460 | $9.626242 \times 10^{-2}$ | $2.131008 \times 10^{-1}$ |
| | 0.900 | 0.307473 | 0.242018 | $6.545548 \times 10^{-2}$ | $2.128819 \times 10^{-1}$ |
| | 0.950 | 0.155653 | 0.122513 | $3.314019 \times 10^{-2}$ | $2.129108 \times 10^{-1}$ |
| | 1.000 | 0.000000 | 0.000000 | $1.001917 \times 10^{-16}$ | $8.222354 \times 10^{-1}$ |

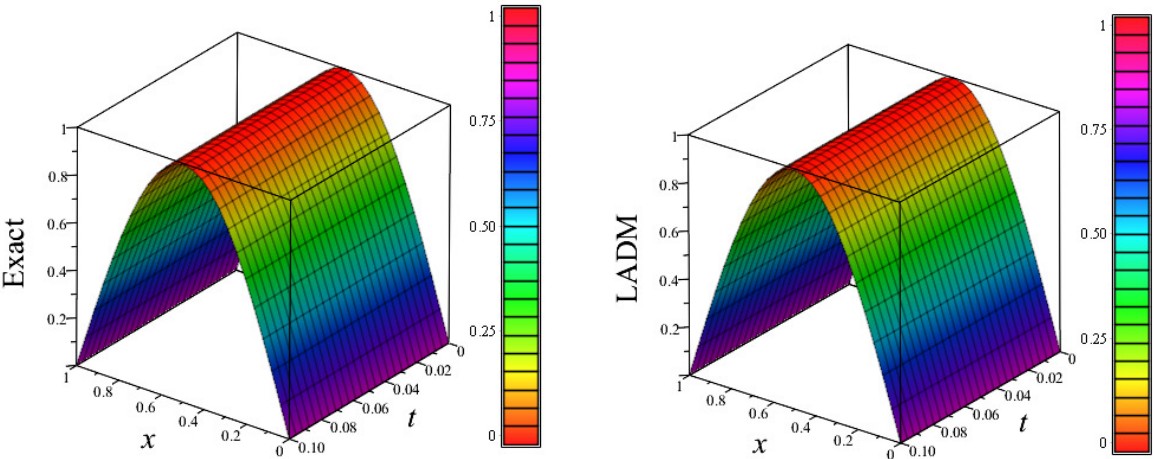

**Figure 6.** 3D plots of exact solution and approximate solution using LADM, vs. $x$ vs. $t$, for numerical experiment 2.

## 3. Homotopy Perturbation Method (HPM)

HPM was proposed by Ji-Huan He in 1999 ([29,30]). HPM is a method not only used for finding accurate asymptotic solutions of nonlinear problem, but also for effectively obtaining a solution in a rapidly convergent series, which is given in a closed-form. This method was applied for PDEs arising in modeling of flow in porous media [42], for the solution of inverse heat conduction problem in [43] and for solving the Volterra–Fredholm integral equations of the second kind in [44]. In this method, the solution is considered to be the summation of an infinite series, which usually converges rapidly to the exact solution.

To illustrate HPM (cf. [29,30]), let us consider the following differential equation:

$$A(u) = f(r), \quad r \in \Omega, \tag{28}$$

supplied with boundary conditions $B(u, \frac{\partial u}{\partial \hat{n}}) = 0$; $r \in \Theta$, where $A$ is a general differential operator, $B$ is a boundary operator, $f(r)$ is a known analytic function, $\Theta$ is the boundary of the domain $\Omega$ and $\frac{\partial u}{\partial \hat{n}}$ denotes directional derivative in the outwarding normal $\hat{n}$ to $\Omega$ [29,30]. Suppose the operator $A$ can be divided into two parts: $M$ and $N$. Therefore, Equation (28) can be rewritten as

$$M(u) + N(u) = f(r). \tag{29}$$

The homotopy $v(r, p) : \Omega \times [0, 1] \to \mathbb{R}$ constructed as follows [29]

$$\mathbb{H}(v, p) = (1 - p)[M(v) - M(y_0)] + p\,[A(v) - f(r)] = 0, \tag{30}$$

where $r \in \Omega$ and $p \in [0, 1]$ is an embedding parameter, and $y_0$ is an initial approximation of (28). Hence, one can see that

$$\mathbb{H}(v, 0) = M(v) - M(y_0) = 0, \quad \mathbb{H}(v, 1) = A(v) - f(r) = 0, \tag{31}$$

and changing the variation of $p$ from 0 to 1 is the same as changing $\mathbb{H}(v, p)$ from $M(v) - M(y_0)$ to $A(v) - f(r)$, which are called homotopic. In topology, this is known as a deformation. Due to the fact that $0 \leq p \leq 1$ can be considered to be a small parameter, by applying the perturbation procedure, the solution of (30) can be expressed as a series in $p$ as

$$v(x, t) = v_0 + p v_1 + p^2 v_2 + p^3 v_3 + \dots \tag{32}$$

By letting $p \to 1$, Equation (30) corresponds to Equation (29), and Equation (32) becomes the approximate solution of Equation (29); that is,

$$u(x,t) = \lim_{p \to 1} v(x,t) = v_0(x,t) + v_1(x,t) + v_2(x,t) + v_3(x,t) + \dots \tag{33}$$

Substituting Equation (32) into Equation (30) and equating the terms with identical powers of $p$, we can obtain

$$\left. \begin{array}{l} p^0 : v_0 - f(x) = 0, \\ p^1 : v_1 - \mathbb{H}(v_0) = 0, \\ \vdots \\ p^n : v_n - \mathbb{H}(v_0, v_1, v_2, \dots, v_{n-1}) = 0, \end{array} \right\} \tag{34}$$

for $n \in \mathbb{N}_0$, where $\mathbb{H}(v_0, v_1, v_2, \dots, v_j)$ depends upon $v_0, v_1, v_2, \dots, v_j$.

**Remark 5.** *Please note that the series in Equation (33) is convergent for most cases; however, the convergent rate relies on the nonlinear operator $A(v)$. Sometimes, even the first approximation is sufficient to obtain the exact solution [29]. As it is shown in [29], the second derivative of $N(v)$ with respect to $v$ must be small, because the parameter $p$ may be relatively large; i.e., $p \to 1$, and the norm of $L^{-1} \partial N / \partial v$ must be smaller than one for the series to converge.*

### 3.1. Solution of Numerical Experiment 1 via HPM

By means of HPM, we construct a homotopy map for the linearized KdV equation as

$$\mathbb{H}(v,p) = (1-p)\left( \frac{\partial v_0}{\partial t} - \frac{\partial u_0}{\partial t} \right) + p\left[ \frac{\partial v}{\partial t} + 2\frac{\partial v}{\partial x} + \frac{\partial^3 v}{\partial x^3} \right] = 0, \tag{35}$$

subject to the initial condition $u(x,0) = \sin(x)$. Substituting Equation (32) into Equation (35), we obtain

$$\left( \frac{\partial v_0}{\partial t} - \frac{\partial u_0}{\partial t} \right) + p\left\{ \frac{\partial v_1}{\partial t} + \left( \frac{\partial u_0}{\partial t} - \frac{\partial v_0}{\partial t} \right) + \left( \frac{\partial v_0}{\partial t} + 2\frac{\partial v_0}{\partial x} + \frac{\partial^3 v_0}{\partial x^3} \right) \right\}$$

$$+ p^2 \left\{ \frac{\partial v_2}{\partial t} - \frac{\partial v_1}{\partial t} + \left( \frac{\partial v_1}{\partial t} + 2\frac{\partial v_1}{\partial x} + \frac{\partial^3 v_1}{\partial x^3} \right) \right\} + \dots = 0.$$

By collecting terms of the same power of $p$, the components of $v_i$'s are given as follows:

$$p^{(0)} : \quad \frac{\partial v_0}{\partial t} - \frac{\partial u_0}{\partial t} = 0, \tag{36a}$$

$$p^{(1)} : \quad \frac{\partial v_1}{\partial t} + \frac{\partial u_0}{\partial t} + 2\frac{\partial v_0}{\partial x} + \frac{\partial^3 v_0}{\partial x^3} = 0, \tag{36b}$$

$$p^{(2)} : \quad \frac{\partial v_2}{\partial t} + 2\frac{\partial v_1}{\partial x} + \frac{\partial^3 v_1}{\partial x^3} = 0, \tag{36c}$$

$$\vdots$$

$$p^{(n)} : \quad \frac{\partial v_n}{\partial t} + 2\frac{\partial v_{n-1}}{\partial x} + \frac{\partial^3 v_{n-1}}{\partial x^3} = 0, \quad n \geq 2. \tag{36d}$$

Thus, solving Equation (36a–d), by using techniques of integration yields

$$v_0 = u_0 = \sin(x), \quad v_1 = -t\cos(x), \quad v_2 = -\frac{t^2}{2!}\sin(x), \ v_3 = \frac{t^3}{3!}\cos(x), \quad v_4 = \frac{t^4}{4!}\sin(x), \quad v_5 = -\frac{t^5}{5!}\cos(x),$$

$$v_6 = -\frac{t^6}{6!}\sin(x), \quad v_7 = \frac{t^7}{7!}\cos(x), \quad v_8 = \frac{t^8}{8!}\sin(x), \ v_9 = -\frac{t^9}{9!}\cos(x), \quad v_{10} = -\frac{t^{10}}{10!}\sin(x),$$

and so on. Thus, the sum of the first ten approximate solutions for Equation (4) is given by

$$u(x,t) = \lim_{k \to 1} v(x,t) = v_0(x,t) + v_1(x,t) + v_2(x,t) + \ldots + v_{10}(x,t)$$

$$= \sin(x) - t\cos(x) - \frac{t^2}{2!}\sin(x) + \frac{t^3}{3!}\cos(x) + \frac{t^4}{4!}\sin(x) - \frac{t^5}{5!}\cos(x)$$

$$- \frac{t^6}{6!}\sin(x) + \frac{t^7}{7!}\cos(x) + \frac{t^8}{8!}\sin(x) - \frac{t^9}{9!}\cos(x) - \frac{t^{10}}{10!}\sin(x). \tag{37}$$

**Remark 6.** *We note that Equation (37) and Equation (21) are exactly the same, which is in full agreement with the result given in [27,45]. Also, the result of approximate solutions for Equation (4) via HPM is simple, quick and easy and it agrees with the approximate solution via LADM given in Equation (20). Table 1 shows both the absolute and relative errors for Equation (4) via HPM as well as LADM.*

### 3.2. Solution of Numerical Experiment 2 via HPM

The inhomogeneous dispersive KdV equation is given by

$$u_t + u_{xxx} = -\sin(\pi x)\sin(t) - \pi^3\cos(\pi x)\cos(t), \quad x \in [0,1], \quad t \in [0, 0.1],$$

subject to the initial condition $u_0(x) = \sin(\pi x)$ [27]. To solve Equation (5) by HPM, we construct the following homotopy:

$$(1-p)\left(\frac{\partial v}{\partial t} - \frac{\partial u_0}{\partial t}\right) + p\left[\frac{\partial v}{\partial t} + \frac{\partial^3 v}{\partial x^3} + \sin(\pi x)\sin(t) + \pi^3\cos(\pi x)\cos(t)\right] = 0. \tag{38}$$

Assuming the solution of Equation (5) takes the form of the series in Equation (32) and substituting this into Equation (38) and collecting terms of the same power of $p$ gives

$$p^{(0)}: \quad \frac{\partial v_0}{\partial t} - \frac{\partial u_0}{\partial t} = 0 \implies v_0 = u_0,$$

$$p^{(1)}: \quad \frac{\partial v_1}{\partial t} = -\frac{\partial u_0}{\partial t} - \frac{\partial^3 v_0}{\partial x^3} - \sin(\pi x)\sin(t) - \pi^3\cos(\pi x)\cos(t),$$

$$\vdots$$

$$p^{(n)}: \quad \frac{\partial v_n}{\partial t} = -\frac{\partial^3 v_{n-1}}{\partial x^3}, \text{ for } n \geq 2.$$

Here, we will use the initial condition $v_0 = u_0 = u(x,0)$ and the first few approximate HPM solutions are given by

$$v_0(x,t) = u(x,0) = \sin(\pi x), \tag{39a}$$

$$v_1(x,t) = -\sin(\pi x) + \sin(\pi x)\cos(t) + \pi^3 t\cos(\pi x) - \pi^3\cos(\pi x)\sin(t), \tag{39b}$$

$$v_2(x,t) = \pi^6\sin(\pi x) - \frac{1}{2}\pi^6 t^2\sin(\pi x) + \sin(t)\pi^3\cos(\pi x) - \pi^3 t\cos(\pi x) - \cos(t)\pi^6\sin(\pi x), \tag{39c}$$

$$v_3(x,t) = -\pi^6\sin(\pi x) + \pi^9 t\cos(\pi x) - \frac{1}{6}\pi^9 t^3\cos(\pi x) + \cos(t)\pi^6\sin(\pi x) + \frac{1}{2}\pi^6 t^2\sin(\pi x) - \sin(t)\pi^9\cos(\pi x), \tag{39d}$$

$$v_4(x,t) = \pi^{12}\sin(\pi x)\left[1 - \frac{t^2}{2!} + \frac{t^4}{4!} - \cos(t)\right] + \pi^9\cos(\pi x)\left[\sin(t) - t + \frac{t^3}{3!}\right], \tag{39e}$$

$$v_5(x,t) = \pi^{12} \sin(\pi x) \left[\cos(t) - t + \frac{t^3}{3!}\right] + \pi^{15} \cos(\pi x) \left[t - \frac{t^3}{3!} + \frac{t^5}{5!} - \sin(t)\right], \tag{39f}$$

$$v_6(x,t) = \pi^{18} \sin(\pi x) \left[1 - \frac{t^2}{2!} + \frac{t^4}{4!} - \frac{t^6}{6!} - \cos(t)\right] - \pi^{15} \cos(\pi x) \left[t - \frac{t^3}{3!} + \frac{t^5}{5!} + \sin(t)\right], \tag{39g}$$

$$v_7(x,t) = -\pi^{18} \sin(\pi x) \left[1 - \frac{t^2}{2!} + \frac{t^4}{4!} - \frac{t^6}{6!} - \cos(t)\right] + \pi^{21} \cos(\pi x) \left[t - \frac{t^3}{3!} - \sin(t) + \frac{t^5}{5!} - \frac{t^7}{7!}\right]. \tag{39h}$$

The sum of the first few approximate solutions using HPM yield

$$S(x,t) = \sum_{i=0}^{7} u_i(x,t) = \sin(\pi x) \cos(t) - \pi^{21} \sin(t) \cos(\pi x) + \pi^{21} \cos(\pi x) \left(t - \frac{t^3}{3!} + \frac{t^5}{5!} - \frac{t^7}{7!}\right) + \dots, \tag{39i}$$

which converges to the exact solution upon Taylor's approximation.

**Remark 7.** *For the inhomogeneous KdV problem, the accuracy of the approximate solution using HPM strongly relies on the initial conditions used to solve this equation. We note that HPM work well for small numerical values of the initial conditions. Once these values are increased, the accuracy of the estimations becomes poor, at least for the number of terms used to approximate the solutions in the present approach (see Table 3 and also Figures 7 and 8). These scenario of convergence of HPM solution from the initial conditions and nature of the source term motivates a thorough investigation on the modification of HPM to obtain an improved result as described below.*

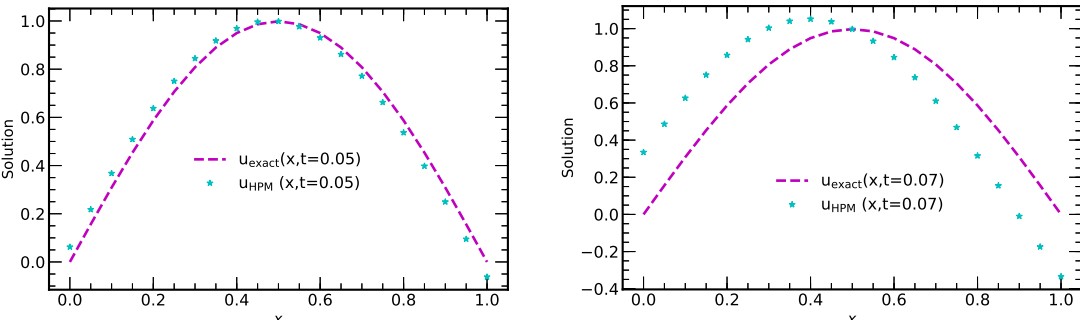

**Figure 7.** Plots of exact solution and approximate solution using HPM at different $t$-values, $t = 0.05, 0.07$ (for numerical experiment 2).

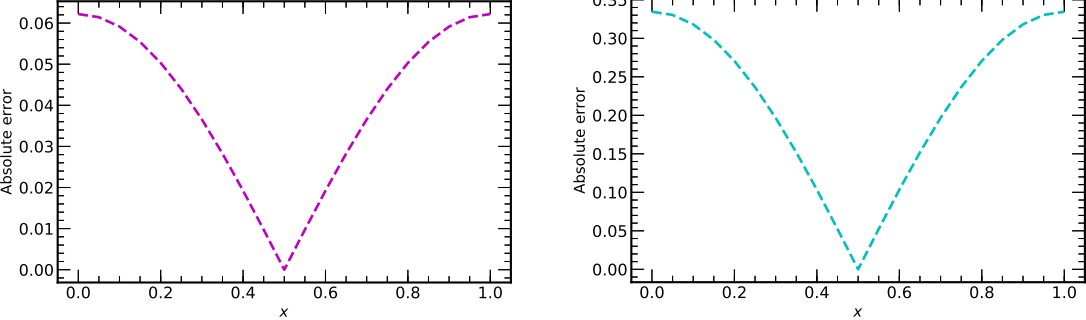

**Figure 8.** Plots of absolute errors vs. $x$ using 7-terms of HPM at different values of time ($t = 0.05, 0.07$).

**Table 3.** Absolute and relative errors at some values of $x$ obtained at times 0.05, 0.07 using 7-terms of HPM (for numerical experiment 2).

| $t$ | Values of $x$ | Exact Solution | Numerical Solution | Absolute Error | Relative Error |
|---|---|---|---|---|---|
| | 0.050 | 0.156239 | 0.217665 | $6.142635 \times 10^{-2}$ | $3.931564 \times 10^{-1}$ |
| | 0.100 | 0.308631 | 0.367779 | $5.914842 \times 10^{-2}$ | $1.916478 \times 10^{-1}$ |
| | 0.150 | 0.453423 | 0.508837 | $5.541345 \times 10^{-2}$ | $1.222113 \times 10^{-1}$ |
| | 0.200 | 0.587051 | 0.637365 | $5.031456 \times 10^{-2}$ | $8.570736 \times 10^{-2}$ |
| | 0.250 | 0.706223 | 0.750199 | $4.397635 \times 10^{-2}$ | $6.226977 \times 10^{-2}$ |
| | 0.300 | 0.808006 | 0.844561 | $3.655552 \times 10^{-2}$ | $4.524164 \times 10^{-2}$ |
| | 0.350 | 0.889893 | 0.918127 | $2.823447 \times 10^{-2}$ | $3.172794 \times 10^{-2}$ |
| | 0.400 | 0.949868 | 0.969086 | $1.921845 \times 10^{-2}$ | $2.023276 \times 10^{-2}$ |
| | 0.450 | 0.986454 | 0.996183 | $9.728888 \times 10^{-3}$ | $9.862485 \times 10^{-3}$ |
| $t = 0.05$ | 0.500 | 0.998750 | 0.998750 | $9.331381 \times 10^{-8}$ | $9.343057 \times 10^{-8}$ |
| | 0.550 | 0.986454 | 0.976725 | $9.728769 \times 10^{-3}$ | $9.862365 \times 10^{-3}$ |
| | 0.600 | 0.949868 | 0.930650 | $1.921821 \times 10^{-2}$ | $2.023251 \times 10^{-2}$ |
| | 0.650 | 0.889893 | 0.861658 | $2.823459 \times 10^{-2}$ | $3.172807 \times 10^{-2}$ |
| | 0.700 | 0.808006 | 0.771450 | $3.655564 \times 10^{-2}$ | $4.524179 \times 10^{-2}$ |
| | 0.750 | 0.706223 | 0.662246 | $4.397659 \times 10^{-2}$ | $6.227011 \times 10^{-2}$ |
| | 0.800 | 0.587051 | 0.536737 | $5.031409 \times 10^{-2}$ | $8.570655 \times 10^{-2}$ |
| | 0.850 | 0.453423 | 0.398010 | $5.541309 \times 10^{-2}$ | $1.222106 \times 10^{-1}$ |
| | 0.900 | 0.308631 | 0.249483 | $5.914819 \times 10^{-2}$ | $1.916471 \times 10^{-1}$ |
| | 0.950 | 0.156239 | 0.094813 | $6.142611 \times 10^{-2}$ | $3.931549 \times 10^{-1}$ |
| | 0.050 | 0.156051 | 0.486414 | $3.303631 \times 10^{-1}$ | $0.2117015 \times 10^{1}$ |
| | 0.100 | 0.308260 | 0.626371 | $3.181107 \times 10^{-1}$ | $0.1031955 \times 10^{1}$ |
| | 0.150 | 0.452879 | 0.750904 | $2.980252 \times 10^{-1}$ | $6.580686 \times 10^{-1}$ |
| | 0.200 | 0.586346 | 0.856947 | $2.706008 \times 10^{-1}$ | $4.615039 \times 10^{-1}$ |
| | 0.250 | 0.705375 | 0.941889 | $2.365139 \times 10^{-1}$ | $3.353024 \times 10^{-1}$ |
| | 0.300 | 0.807036 | 1.003639 | $1.966030 \times 10^{-1}$ | $2.436113 \times 10^{-1}$ |
| | 0.350 | 0.888824 | 1.040676 | $1.518513 \times 10^{-1}$ | $1.708451 \times 10^{-1}$ |
| | 0.400 | 0.948727 | 1.052088 | $1.033602 \times 10^{-1}$ | $1.089462 \times 10^{-1}$ |
| | 0.450 | 0.985269 | 1.037594 | $5.232441 \times 10^{-2}$ | $5.310670 \times 10^{-2}$ |
| $t = 0.07$ | 0.500 | 0.997551 | 0.997551 | $3.354171 \times 10^{-9}$ | $3.362405 \times 10^{-9}$ |
| | 0.550 | 0.985269 | 0.932945 | $5.232417 \times 10^{-2}$ | $5.310645 \times 10^{-2}$ |
| | 0.600 | 0.948727 | 0.845367 | $1.033602 \times 10^{-1}$ | $1.089462 \times 10^{-1}$ |
| | 0.650 | 0.888824 | 0.736973 | $1.518512 \times 10^{-1}$ | $1.708450 \times 10^{-1}$ |
| | 0.700 | 0.807036 | 0.610433 | $1.966030 \times 10^{-1}$ | $2.436113 \times 10^{-1}$ |
| | 0.750 | 0.705375 | 0.468861 | $2.365137 \times 10^{-1}$ | $3.353020 \times 10^{-1}$ |
| | 0.800 | 0.586346 | 0.315745 | $2.706009 \times 10^{-1}$ | $4.615041 \times 10^{-1}$ |
| | 0.850 | 0.452879 | 0.154854 | $2.980249 \times 10^{-1}$ | $6.580678 \times 10^{-1}$ |
| | 0.900 | 0.308260 | −0.009850 | $3.181102 \times 10^{-1}$ | $0.1031954 \times 10^{1}$ |
| | 0.950 | 0.156051 | −0.174312 | $3.303633 \times 10^{-1}$ | $0.2117016 \times 10^{1}$ |

*3.3. Homotopy Perturbation Transform Method (HPTM)*

This method combines the Homotopy Perturbation Method with Laplace transform for solving various types of linear and nonlinear PDEs [46,47]. To illustrate the basic idea of HPTM, we consider

$$\mathsf{L}_t u(x,t) + Ru(x,t) + Nu(x,t) = g(x,t), \tag{40}$$

with an initial condition $u(x,0) = h(x)$, where $\mathsf{L}_t = \frac{\partial}{\partial t}$, $R$ is a linear operator that includes partial derivatives with respect to $x$, $N$ is a nonlinear operator and $g$ is a non-homogeneous term, which is $u$-independent. Taking the Laplace transform on both sides of Equation (40), we get

$$\hat{\mathcal{L}}\left\{\mathsf{L}_t u(x,t)\right\} = \hat{\mathcal{L}}\left\{g(x,t) - Ru(x,t) - Nu(x,t)\right\}. \tag{41}$$

From the differentiation property of Laplace transform $\hat{\mathcal{L}}$ of Equation (41), we obtain

$$\hat{\mathcal{L}}\left[u(x,t)\right] = \frac{h(x)}{s} + \frac{1}{s}\hat{\mathcal{L}}[g(x,t)] - \frac{1}{s}\left[Ru(x,t) + Nu(x,t)\right]. \tag{42}$$

Operating the Laplace inverse of Equation (42) yields

$$u(x,t) = h(x) + \hat{\mathcal{L}}^{-1}\left[\frac{1}{s}\hat{\mathcal{L}}[g(x,t)]\right] - \hat{\mathcal{L}}^{-1}\left[\frac{1}{s}\hat{\mathcal{L}}[Ru(x,t) + Nu(x,t)]\right]. \tag{43}$$

Now applying the HPM series

$$u(x,t) = \sum_{n=0}^{\infty} p^n u_n(x,t), \tag{44}$$

and the nonlinear term can be decomposed as

$$Nu(x,t) = \sum_{n=0}^{\infty} p^n \mathbf{H}_n(u), \tag{45}$$

for some He's polynomials (see [29,30])

$$\mathbf{H}_n(u_0, u_1, \ldots, u_n) = \frac{1}{n!}\frac{\partial^n}{\partial p^n}\left[N\left(\sum_{i=0}^{\infty} p^i u_i\right)\right]_{p=0}, \quad n = 0, 1, 2, \ldots.$$

Substituting Equations (44) and (45) in Equation (43), we get

$$\sum_{n=0}^{\infty} p^n u_n(x,t) = h(x) + \hat{\mathcal{L}}^{-1}\left[\frac{1}{s}\hat{\mathcal{L}}[g(x,t)]\right] - \hat{\mathcal{L}}^{-1}\left[\frac{1}{s}\hat{\mathcal{L}}\left[R\sum_{n=0}^{\infty} p^n u_n(x,t) + \sum_{n=0}^{\infty} p^n \mathbf{H}_n(u)\right]\right], \tag{46}$$

which is the coupling of the Laplace transform and the homotopy perturbation method using He's polynomials. Comparing the coefficient of like powers of $p$, the following approximations are obtained.

$$u_0(x,t) = h(x) + \hat{\mathcal{L}}^{-1}\left[\frac{1}{s}\hat{\mathcal{L}}[g(x,t)]\right], \tag{47a}$$

$$u_1(x,t) = -\hat{\mathcal{L}}^{-1}\left[\frac{1}{s}\hat{\mathcal{L}}[Ru_0(x,t) + \mathbf{H}_0(u)]\right], \tag{47b}$$

$$u_2(x,t) = -\hat{\mathcal{L}}^{-1}\left[\frac{1}{s}\hat{\mathcal{L}}[Ru_1(x,t) + \mathbf{H}_1(u)]\right], \tag{47c}$$

$$\vdots$$

$$u_n(x,t) = -\hat{\mathcal{L}}^{-1}\left[\frac{1}{s}\hat{\mathcal{L}}[Ru_n(x,t) + \mathbf{H}_n(u)]\right], \quad n \geq 2. \tag{47d}$$

### 3.4. Application of HPTM to Equation (5)

Consider the inhomogeneous KdV equation given in Equation (5). By applying the aforementioned method, HPTM, we obtain

$$\sum_{n=0}^{\infty} p^n u_n(x,t) = h(x) - \hat{\mathcal{L}}^{-1}\left[\frac{1}{s}\hat{\mathcal{L}}[\sin(\pi x)\sin(t) + \pi^3\cos(\pi x)\cos(t)]\right] - \hat{\mathcal{L}}^{-1}\left[\frac{1}{s}\hat{\mathcal{L}}\left[\sum_{n=0}^{\infty} p^n \frac{\partial^3 u_n(x,t)}{\partial x^3}\right]\right]. \tag{48}$$

Comparing the coefficients of various powers of $p$ in Equation (48), we get

$$u_0(x,t) = h(x) + \hat{\mathcal{L}}^{-1}\left[\frac{1}{s}\hat{\mathcal{L}}[g(x,t)]\right] = \sin(\pi x)\cos(t) - \pi^3\cos(\pi x)\sin(t),$$

$$u_1(x,t) = -\hat{\mathcal{L}}^{-1}\left[\frac{1}{s}\hat{\mathcal{L}}\left[\frac{\partial^3 u_0(x,t)}{\partial x^3}\right]\right] = \pi^3\cos(\pi x)\sin(t) - \pi^6\sin(\pi x)\cos(t) + \pi^6\sin(\pi x), \quad (49)$$

$$u_2(x,t) = -\hat{\mathcal{L}}^{-1}\left[\frac{1}{s}\hat{\mathcal{L}}\left[\frac{\partial^3 u_1(x,t)}{\partial x^3}\right]\right] = \pi^9\cos(\pi x)[\sin(t) - t] + \pi^{12}\sin(\pi x)\left[1 - \cos(t) - \frac{t^2}{2!}\right].$$

This procedure proceeds in a similar manner for higher order iteration of approximate solution with some of the self-cancelling terms to obtain the exact solution in Equation (23).

**Remark 8.** *HPTM can be perceived as a good refinement of HPM and the results obtained by HPTM coincide with LADM for the inhomogeneous dispersive KdV equation given by Equation (5). See Figure 9 for the graphical representation of exact and HPTM solution for Equation (5).*

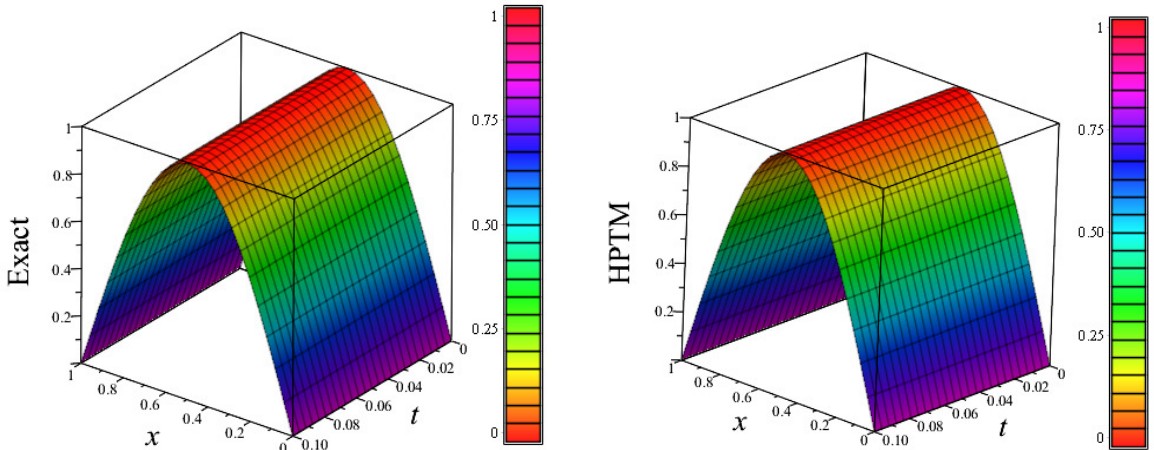

**Figure 9.** 3D plots of exact solution and approximate solution using HPTM, vs. $x$ vs. $t$, for numerical experiment 2.

### 3.5. HPM with a Modified Homotopy

For the inhomogeneous dispersive KdV equation, the numerical result obtained in Table 3 can be improved by constructing a modified form of new homotopy, $v(r,p) : \Omega \times [0,1] \to \mathbb{R}$, which given by

$$\mathbb{H}(v,p) = (1-p)[M(v) - M(y_0)] + p[A(v)] - f(r) = 0, \quad (50)$$

that specifically takes the form

$$(1-p)\left(\frac{\partial v}{\partial t} - \frac{\partial u_0}{\partial t}\right) + p\left[\frac{\partial v}{\partial t} + \frac{\partial^3 v}{\partial x^3}\right] + \sin(\pi x)\sin(t) + \pi^3\cos(\pi x)\cos(t) = 0. \quad (51)$$

We note here that we made some modification by shifting the source term to the standard HPM, as shown in Equation (50), for enhancing the convergence of the series solution. This modification allows suitable choice of the initial approximation that makes the approximate solutions to rapidly converge to the exact solution. One can check that the obtained results, using Equation (51), for the modified HPM are in full agreement with the approximate solutions obtained using LADM. Just to mention the first few approximate solutions obtained via modified HPM as

$$
\begin{aligned}
v_0(x,t) &= u(x,0) + \int_0^t g(x,s)\, ds = \sin(\pi x) - \int_0^t \left( \sin(\pi x)\sin(s) + \pi^3 \cos(\pi x)\cos(s) \right) ds \\
&= -\pi^3 \cos(\pi x)\sin(t) + \sin(\pi x)\cos(t), \\
v_1(x,t) &= -\int_0^t v_{0,xxx}(x,s)\, ds = \pi^3 \cos(\pi x)\sin(t) - \pi^6 \sin(\pi x)\cos(t) + \pi^6 \sin(\pi x), \\
v_2(x,t) &= -\int_0^t v_{1,xxx}(x,s)\, ds = \pi^9 t \cos(\pi x) - \pi^9 \cos(\pi x)\sin(t) + \pi^6 \sin(\pi x)\cos(t) - \pi^6 \sin(\pi x),
\end{aligned}
\tag{52}
$$

and, for the higher order approximate solution, we have $v_n(x,t) = -\int_0^t v_{n-1,xxx}(x,s)\, ds, \quad n \geq 3$.

**Remark 9.** *We note that the modified HPM using the newly constructed homotopy, HPTM and LADM are equivalent approximation schemes when applied to the linearized in-homogeneous Equation given in Equation (5). The absolute and relative errors are given in Table 2. We note that HPTM and LADM are equivalent methods for numerical experiment 2.*

## 4. Reduced Differential Transform Method (RDTM)

Zhao [48] introduced differential transform method (DTM) to solve PDEs involved in electric circuit problems. DTM involves Taylor series expansion, which gives a polynomial series solution via an iterative procedure. Reduced differential transform method (RDTM) is very powerful method to obtain analytical approximate solutions to linear and nonlinear ordinary differential equations [31,32] and for systems of differential equations [49]. Basic definitions and properties for RDTM can be found in [31,32,50].

**Definition 1.** *Consider a function of $n+1$ variables. The reduced differential transform of $u(\tilde{X},t) = u(x_1, x_2, \ldots, x_n, t)$ (where $\tilde{X} \in \mathbb{R}^n$) with respect to t is defined by*

$$
U_k(\tilde{X}) = \frac{1}{k!} \left[ \frac{\partial^k}{\partial t^k} u(\tilde{X},t) \right]_{t=0}, \quad k = 0,1,2,\ldots,
\tag{53}
$$

*where $U_k(\tilde{X})$ denotes the transform function of $u(\tilde{X},t)$.*

**Definition 2.** *The differential inverse transform of $\{U_k(\tilde{X})\}_{k=0}^n$ is defined by*

$$
u(\tilde{X},t) = \sum_{k=0}^{\infty} U_k(\tilde{X})\, t^k.
\tag{54}
$$

By substituting Equation (53) into Equation (54), we obtain

$$
u(\tilde{X},t) = \sum_{k=0}^{\infty} \frac{1}{k!} \left[ \frac{\partial^k}{\partial t^k} u(\tilde{X},t) \right]_{t=0} t^k.
$$

From the above definitions, we see that RDTM is obtained from power series expansion. Please note that RDTM is close to the one dimensional DTM because RDTM is considered to be the standard DTM of $u(\tilde{X},t)$ with respect to the variable $t$. However, the corresponding recursive algebraic equation is the function of the variable $\tilde{X} = (x_1, x_2, \ldots, x_n)$.

The fundamental mathematical operations for RDTM [31,32] are listed in Table 4.

**Table 4.** Transformed functions using RTDM.

| Function $f(\tilde{X}, t)$ | Transformed Function $F_k(\tilde{X})$ |
|---|---|
| $au(\tilde{X}, t) \pm bu(\tilde{X}, t)$ | $aU_k(\tilde{X}) \pm bV_k(\tilde{X})$ |
| $u(\tilde{X}, t) \cdot v(\tilde{X}, t)$ | $\displaystyle\sum_{i=0}^{k} U_i(\tilde{X}) \cdot V_{k-i}(\tilde{X})$ |
| $\dfrac{\partial^n}{\partial t^n} u(\tilde{X}, t)$ | $\dfrac{(k+n)!}{k!} U_{k+n}(\tilde{X})$ |
| $\dfrac{\partial^n}{\partial x_i^n} u(\tilde{X}, t)$ | $\dfrac{\partial^n U_k(\tilde{X})}{\partial x_i}$ |
| $x^m t^n u(\tilde{X}, t)$ | $\tilde{X}^{\bar{m}} t^n U_{k-n}(\tilde{X}) \ (\text{where } \tilde{X}^{\bar{m}} = x_1^{m_1} x_2^{m_2} \dots x_n^{m_n})$ |
| $\sin\left(\alpha x + \beta y + \gamma z + w\, t\right)$ | $\dfrac{w^k}{k!} \cdot \sin\left(\dfrac{k\pi}{2!} + \alpha x + \beta y + \gamma z\right)$ |
| $\cos\left(\alpha x + \beta y + \gamma z + w\, t\right)$ | $\dfrac{w^k}{k!} \cdot \cos\left(\dfrac{k\pi}{2!} + \alpha x + \beta y + \gamma z\right)$ |

*4.1. Solution of Numerical Experiment 1 via RDTM*

Consider the homogeneous KdV equation given in Equation (4) with initial condition in (16). Applying RDTM to Equations (4) and (16), we obtain the recursive relation

$$(k+1)\, U_{k+1}(x) + 2\frac{\partial U_k(x)}{\partial x} + \frac{\partial^3 U_k(x)}{\partial x^3} = 0, \tag{55}$$

where $U_k(x)$ is the transform function in the $t$-dimensional spectrum. We see that

$$u_0(x) = \sin(x). \tag{56}$$

Substituting the initial condition (56) into Equation (55), we obtain the following approximations successively

$$U_1(x) = -\cos(x), \quad U_2(x) = -\frac{1}{2}\sin(x),$$

$$U_3(x) = \frac{1}{3!}\cos(x), \quad U_4(x) = \frac{1}{4!}\sin(x),$$

$$U_5(x) = -\frac{1}{5!}\cos(x), \quad U_6(x) = -\frac{1}{6!}\sin(x),$$

$$U_7(x) = \frac{1}{7!}\cos(x), \quad U_8(x) = \frac{1}{8!}\sin(x),$$

$$U_9(x) = -\frac{1}{9!}\cos(x), \quad U_{10}(x) = -\frac{1}{10!}\sin(x),$$

and so on after many iterations, we have

$$U_k(x) = \begin{cases} \dfrac{(-1)^{\left\lfloor \frac{(k-1)}{2} \right\rfloor + 1}}{k!}\cos(x), & \text{for } k \text{ is odd,} \\[4mm] \dfrac{(-1)^{\left\lfloor \frac{(k-1)}{2} \right\rfloor}}{k!}\sin(x), & \text{for } k \text{ is even,} \end{cases}$$

and applying the differential-inverse transform $\{U_k(x)\}_{k \geq 0}$ gives the following approximate solution

$$u(x,t) = \sum_{k\geq 0} U_k(x)\, t^k = \sum_{k\geq 0} U_{2k}(x)\, t^{2k} + \sum_{k\geq 0} U_{2k+1}(x)\, t^{2k+1},$$

$$= \sin(x)\left[1 - \frac{t^2}{2!} + \frac{t^4}{4!} - \frac{t^6}{6!} + \frac{t^8}{8!} + \cdots\right] - \cos(x)\left[t - \frac{t^3}{3!} + \frac{t^5}{5!} - \frac{t^7}{7!} + \cdots\right],$$

$$= \sin(x)\cos(t) - \cos(x)\sin(t) = \sin(x-t),$$

which coincides with the exact solution of Equation (4).

**Remark 10.** *The approximate solutions for Equation (4) via RDTM is quick and very effective and it agrees with the approximate solution via LADM as well as HPM. We also note that Table 1 shows both the absolute and relative error results for Equation (4) via RDTM as well as LADM/HPM.*

*4.2. Solution of Numerical Experiment 2 via RDTM*

By considering the inhomogeneous KdV equation given by Equation (5) with the initial condition in (23), we apply RDTM to obtain following recursive equation:

$$U_{k+1}(x) = \frac{-1}{k+1}\left\{\frac{\partial^3 U_k(x)}{\partial x^3} - \sin(\pi x)\left[\frac{1}{k!}\sin\left(\frac{k\pi}{2}\right)\right] - \pi^3\cos(\pi x)\left[\frac{1}{k!}\cos\left(\frac{k\pi}{2}\right)\right]\right\}, \tag{57}$$

From Equation (53), the initial condition $u(x,0) = \sin(\pi x)$ can be transformed at $t = 0$ as

$$u_0(x) = \sin(\pi x), \tag{58}$$

where $U_k(x)$ is the transform function in the $t$-dimensional spectrum. Substituting the transformed condition Equation (58) into Equation (57), we obtain the following approximations

$$u_1(x) = 0, \quad U_2(x) = -\frac{1}{2!}\sin(\pi x), \quad U_3(x) = 0, \quad U_4(x) = \frac{1}{4!}\sin(\pi x), \quad U_5(x) = 0,$$

$$U_6(x) = -\frac{1}{6!}\sin(\pi x), \quad U_7(x) = 0, \quad U_8(x) = \frac{1}{8!}\sin(\pi x), \quad U_9(x) = 0, \quad U_{10}(x) = -\frac{1}{10!}\sin(\pi x),$$

and so on, after many iterations, we have

$$U_k(x) = \begin{cases} \dfrac{(-1)^{\lfloor \frac{k}{2}\rfloor}}{k!}\sin(\pi x), & \text{for } k \text{ is even,} \\ 0, & \text{for } k \text{ is odd.} \end{cases}$$

Then, using the inverse transformation Equation (54) $\{U_k(x)\}_{k\geq 0}$ will provide the following approximate solution

$$u(x,t) = \sum_{k\geq 0} U_k(x)\, t^k = \sin(\pi x)\left\{1 - \frac{t^2}{2!} + \frac{t^4}{4!} - \frac{t^6}{6!} + \frac{t^8}{8!} - \frac{t^{10}}{10!} + \cdots\right\},$$

which coincides with the exact solution of Equation (5).

**Remark 11.** *Table 5 demonstrates the exact and approximate solutions obtained by RDTM with absolute and relative errors revealing that the method is very effective and convenient. This fact is also shown in Figures 10–12. For the inhomogeneous dispersive KdV equation, RDTM is more precise than LADM. HPM is the least performing scheme as it highly relies on the initial guess; however, by using the modified homotopy in (50), we obtained a good result for the inhomogeneous KdV equation via modified HPM. We can see from the above that the approximate solutions obtained by RDTM converge quickly to the exact solution at about the seventh iteration compared to the other semianalytic methods.*

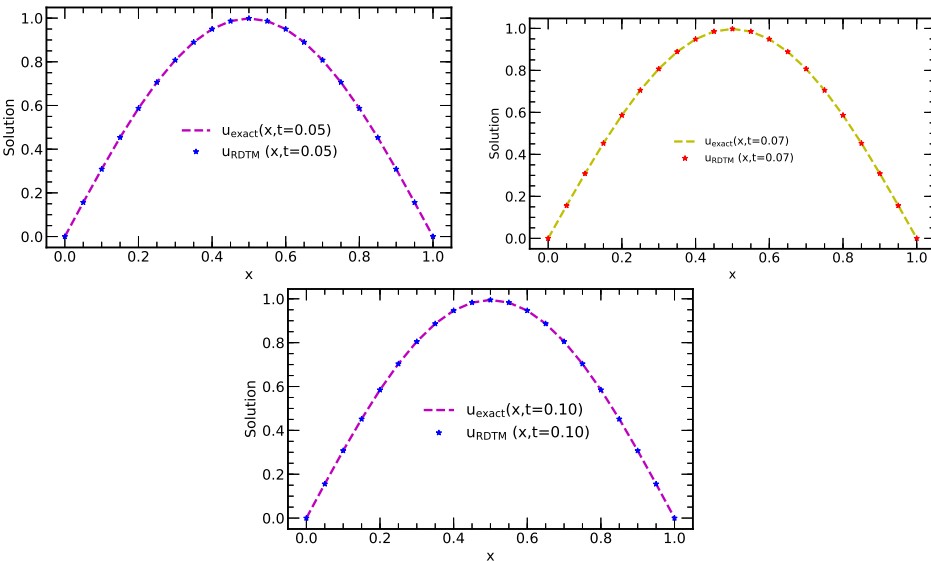

**Figure 10.** Plots of exact solution and approximate solution using RDTM at $t = 0.05, 0.07, 0.10$ (for numerical experiment 2).

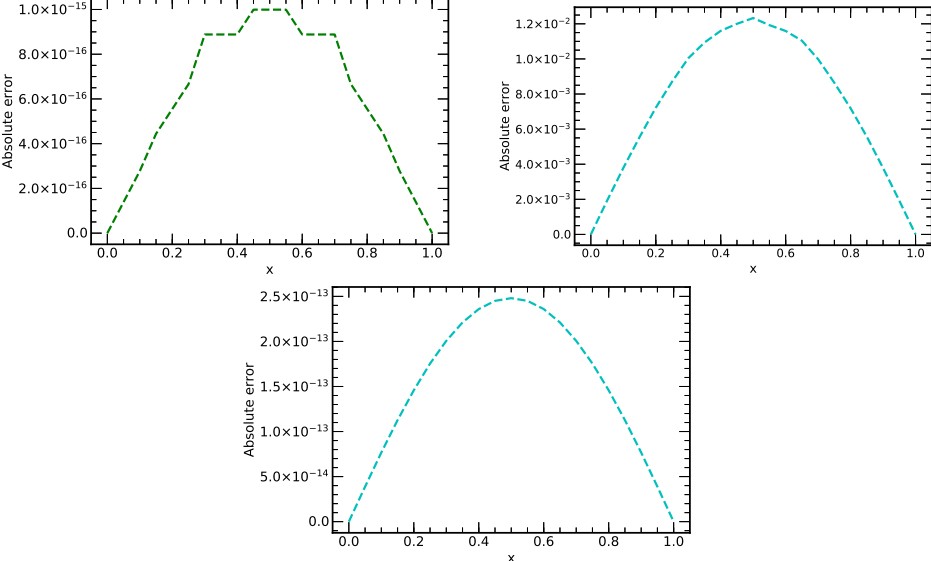

**Figure 11.** Plots for absolute errors at different values of $t$ ($t = 0.05, 0.07, 0.10$).

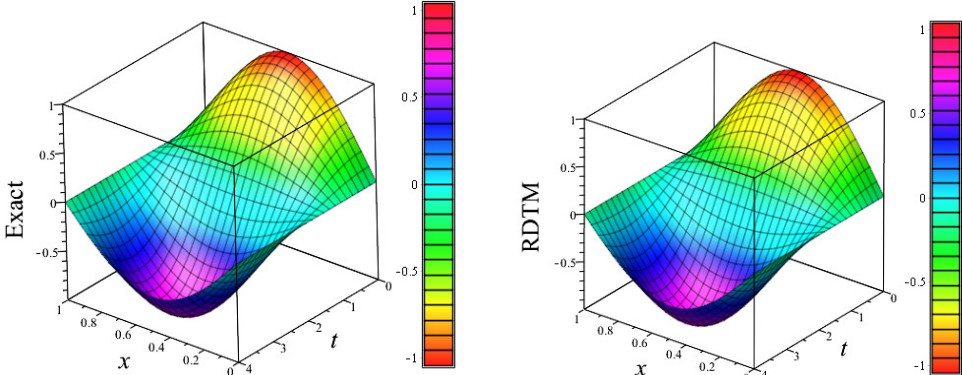

**Figure 12.** 3D plots of exact and approximate solution using RDTM, vs. x vs. t, for numerical experiment 2.

**Table 5.** Absolute and relative errors at some values of $x$ obtained at times 0.05, 0.07, 0.10 using 7-terms of RDTM (for numerical experiment 2).

| $t$ | Values of $x$ | Exact Solution | Numerical Solution | Absolute Error | Relative Error |
|---|---|---|---|---|---|
| | 0.050 | 0.156239 | 0.156239 | $1.387779 \times 10^{-16}$ | $8.882412 \times 10^{-16}$ |
| | 0.100 | 0.308631 | 0.308631 | $2.775558 \times 10^{-16}$ | $8.993132 \times 10^{-16}$ |
| | 0.150 | 0.453423 | 0.453423 | $4.440892 \times 10^{-16}$ | $9.794145 \times 10^{-16}$ |
| | 0.200 | 0.587051 | 0.587051 | $5.551115 \times 10^{-16}$ | $9.455939 \times 10^{-16}$ |
| | 0.250 | 0.706223 | 0.706223 | $6.661338 \times 10^{-16}$ | $9.432343 \times 10^{-16}$ |
| | 0.300 | 0.808006 | 0.808006 | $8.881784 \times 10^{-16}$ | $1.099223 \times 10^{-15}$ |
| | 0.350 | 0.889893 | 0.889893 | $8.881784 \times 10^{-16}$ | $9.980733 \times 10^{-16}$ |
| | 0.400 | 0.949868 | 0.949868 | $8.881784 \times 10^{-16}$ | $9.350546 \times 10^{-16}$ |
| | 0.450 | 0.986454 | 0.986454 | $9.992007 \times 10^{-16}$ | $1.012922 \times 10^{-15}$ |
| | 0.500 | 0.998750 | 0.998750 | $9.992007 \times 10^{-16}$ | $1.000451 \times 10^{-15}$ |
| $t = 0.05$ | 0.550 | 0.986454 | 0.986454 | $9.992007 \times 10^{-16}$ | $1.012922 \times 10^{-15}$ |
| | 0.600 | 0.949868 | 0.949868 | $8.881784 \times 10^{-16}$ | $9.350546 \times 10^{-16}$ |
| | 0.650 | 0.889893 | 0.889893 | $8.881784 \times 10^{-16}$ | $9.980733 \times 10^{-16}$ |
| | 0.700 | 0.808006 | 0.808006 | $8.881784 \times 10^{-16}$ | $1.099223 \times 10^{-15}$ |
| | 0.750 | 0.706223 | 0.706223 | $6.661338 \times 10^{-16}$ | $9.432343 \times 10^{-16}$ |
| | 0.800 | 0.587051 | 0.587051 | $5.551115 \times 10^{-16}$ | $9.455939 \times 10^{-16}$ |
| | 0.850 | 0.453423 | 0.453423 | $4.440892 \times 10^{-16}$ | $9.794145 \times 10^{-16}$ |
| | 0.900 | 0.308631 | 0.308631 | $2.775558 \times 10^{-16}$ | $8.993132 \times 10^{-16}$ |
| | 0.950 | 0.156239 | 0.156239 | $1.387779 \times 10^{-16}$ | $8.882412 \times 10^{-16}$ |
| | 1.000 | 0.000000 | 0.000000 | $1.232595 \times 10^{-31}$ | $1.007750 \times 10^{-15}$ |
| | 0.050 | 0.156051 | 0.156051 | $2.248202 \times 10^{-15}$ | $1.440681 \times 10^{-14}$ |
| | 0.100 | 0.308260 | 0.308260 | $4.385381 \times 10^{-15}$ | $1.422623 \times 10^{-14}$ |
| | 0.150 | 0.452879 | 0.452879 | $6.550316 \times 10^{-15}$ | $1.446373 \times 10^{-14}$ |
| | 0.200 | 0.586346 | 0.586346 | $8.437695 \times 10^{-15}$ | $1.439031 \times 10^{-14}$ |
| | 0.250 | 0.705375 | 0.705375 | $1.010303 \times 10^{-14}$ | $1.432292 \times 10^{-14}$ |
| | 0.300 | 0.807036 | 0.807036 | $1.165734 \times 10^{-14}$ | $1.444464 \times 10^{-14}$ |
| | 0.350 | 0.888824 | 0.888824 | $1.276756 \times 10^{-14}$ | $1.436455 \times 10^{-14}$ |
| | 0.400 | 0.948727 | 0.948727 | $1.365574 \times 10^{-14}$ | $1.439375 \times 10^{-14}$ |
| | 0.450 | 0.985269 | 0.985269 | $1.409983 \times 10^{-14}$ | $1.431064 \times 10^{-14}$ |
| $t = 0.07$ | 0.500 | 0.997551 | 0.997551 | $1.432188 \times 10^{-14}$ | $1.435704 \times 10^{-14}$ |
| | 0.550 | 0.985269 | 0.985269 | $1.409983 \times 10^{-14}$ | $1.431064 \times 10^{-14}$ |
| | 0.600 | 0.948727 | 0.948727 | $1.365574 \times 10^{-14}$ | $1.439375 \times 10^{-14}$ |
| | 0.650 | 0.888824 | 0.888824 | $1.276756 \times 10^{-14}$ | $1.436455 \times 10^{-14}$ |
| | 0.700 | 0.807036 | 0.807036 | $1.165734 \times 10^{-14}$ | $1.444464 \times 10^{-14}$ |
| | 0.750 | 0.705375 | 0.705375 | $1.010303 \times 10^{-14}$ | $1.432292 \times 10^{-14}$ |
| | 0.800 | 0.586346 | 0.586346 | $8.437695 \times 10^{-15}$ | $1.439031 \times 10^{-14}$ |
| | 0.850 | 0.452879 | 0.452879 | $6.550316 \times 10^{-15}$ | $1.446373 \times 10^{-14}$ |
| | 0.900 | 0.308260 | 0.308260 | $4.385381 \times 10^{-15}$ | $1.422623 \times 10^{-14}$ |
| | 0.950 | 0.156051 | 0.156051 | $2.248202 \times 10^{-15}$ | $1.440681 \times 10^{-14}$ |
| | 1.000 | 0.000000 | 0.000000 | $1.750285 \times 10^{-30}$ | $1.432725 \times 10^{-14}$ |
| | 0.050 | 0.155653 | 0.155653 | $3.880229 \times 10^{-14}$ | $2.492873 \times 10^{-13}$ |
| | 0.100 | 0.307473 | 0.307473 | $7.666090 \times 10^{-14}$ | $2.493255 \times 10^{-13}$ |
| | 0.150 | 0.451722 | 0.451722 | $1.126321 \times 10^{-13}$ | $2.493392 \times 10^{-13}$ |
| | 0.200 | 0.584849 | 0.584849 | $1.457723 \times 10^{-13}$ | $2.492478 \times 10^{-13}$ |
| | 0.250 | 0.703574 | 0.703574 | $1.754152 \times 10^{-13}$ | $2.493202 \times 10^{-13}$ |
| | 0.300 | 0.804975 | 0.804975 | $2.006173 \times 10^{-13}$ | $2.492217 \times 10^{-13}$ |
| | 0.350 | 0.886555 | 0.886555 | $2.210454 \times 10^{-13}$ | $2.493307 \times 10^{-13}$ |
| | 0.400 | 0.946305 | 0.946305 | $2.359224 \times 10^{-13}$ | $2.493090 \times 10^{-13}$ |
| | 0.450 | 0.982754 | 0.982754 | $2.450262 \times 10^{-13}$ | $2.493261 \times 10^{-13}$ |
| $t = 0.10$ | 0.500 | 0.995004 | 0.995004 | $2.480238 \times 10^{-13}$ | $2.492691 \times 10^{-13}$ |
| | 0.550 | 0.982754 | 0.982754 | $2.450262 \times 10^{-13}$ | $2.493261 \times 10^{-13}$ |
| | 0.600 | 0.946305 | 0.946305 | $2.359224 \times 10^{-13}$ | $2.493090 \times 10^{-13}$ |
| | 0.650 | 0.886555 | 0.886555 | $2.210454 \times 10^{-13}$ | $2.493307 \times 10^{-13}$ |
| | 0.700 | 0.804975 | 0.804975 | $2.006173 \times 10^{-13}$ | $2.492217 \times 10^{-13}$ |
| | 0.750 | 0.703574 | 0.703574 | $1.754152 \times 10^{-13}$ | $2.493202 \times 10^{-13}$ |
| | 0.800 | 0.584849 | 0.584849 | $1.457723 \times 10^{-13}$ | $2.492478 \times 10^{-13}$ |
| | 0.850 | 0.451722 | 0.451722 | $1.126321 \times 10^{-13}$ | $2.493392 \times 10^{-13}$ |
| | 0.900 | 0.307473 | 0.307473 | $7.666090 \times 10^{-14}$ | $2.493255 \times 10^{-13}$ |
| | 0.950 | 0.155653 | 0.155653 | $3.880229 \times 10^{-14}$ | $2.492873 \times 10^{-13}$ |
| | 1.000 | 0.000000 | 0.000000 | $3.037114 \times 10^{-29}$ | $2.492444 \times 10^{-13}$ |

## 5. Application of RDTM to the 3D Linearized KdV Equation

We now consider a three-dimensional inhomogeneous dispersive KdV equation [27]

$$
u_t + u_{xxx} + u_{yyy} + u_{zzz} = -3\cos(x + 2y + 3z)\sin(t) + \sin(x + 2y + 3z)\cos(t),
$$
$$
\text{where} \quad 0 \le x, y, z \le 4.0, \quad t \in [0, 1.0],
$$
(59)

subject to the initial condition

$$
u(x, y, z, 0) = 0,
$$
(60)

and the time-dependent boundary conditions are assumed to be prescribed [27,45]. Exact solution for Equation (59) is

$$
u(x, y, z, t) = \sin(x + 2y + 3z) \; \sin(t).
$$
(61)

Some of the methods used for semi-analytic solution of Equation (59) are Adomian decomposition method and Variational iteration method [45]. We now propose reduced differential transform method (RDTM) to solve the dispersive KdV equation given in Equation (59). By comparing with LADM, RDTM can be applied directly to solve the problem without using Adomian polynomials.

We now show the applicability and efficiency of RDTM for solving Equation (59). Applying RDTM for Equations (59) and (60) and using Table 4, we obtain the following recursive equation

$$
U_{k+1}(x, y, z) = -\frac{1}{k+1}\left\{ \frac{\partial^3 U_k(x, y, z)}{\partial x^3} + \frac{\partial^3 U_k(x, y, z)}{\partial y^3} + \frac{\partial^3 U_k(x, y, z)}{\partial z^3} - 3\cos(x + 2y + 3z)\left[\frac{1}{k!}\sin\left(\frac{k\pi}{2}\right)\right] \right.
$$
$$
\left. + \sin(x + 2y + 3z)\left[\frac{1}{k!}\cos\left(\frac{k\pi}{2}\right)\right] \right\},
$$
(62)

From Equation (53), the initial conditions given in Equation (60) can be transformed at $t = 0$ as

$$
U_0(x, y, z) = 0.
$$
(63)

Substituting the transformed condition Equation (63) into Equation (62) and by straightforward iterative steps, the following $U_k(x, y, z)$, $k = 0, 1, 2, \ldots, n$ values are obtained:

$$
U_1(x, y, z) = \sin(x + 2y + 3z), \quad U_2(x, y, z) = 0, \quad U_3(x, y, z) = -\frac{1}{3!}\sin(x + 2y + 3z),
$$

$$
U_4(x, y, z) = 0, \quad U_5(x, y, z) = \frac{1}{5!}\sin(x + 2y + 3z), \quad U_6(x, y, z) = 0, \quad U_7(x, y, z) = -\frac{1}{7!}\sin(x + 2y + 3z),
$$

$$
U_8(x, y, z) = 0, \quad U_9(x, y, z) = \frac{1}{9!}\sin(x + 2y + 3z), \quad U_{10}(x, y, z) = 0.
$$

Concise formulation, after many iterations, takes the form

$$
U_k(x, y, z) = \begin{cases} \dfrac{(-1)^k}{(2k+1)!} \; \sin(x + 2y + 3z), & \text{for } k \text{ is odd,} \\ 0, & \text{for } k \text{ is even.} \end{cases}
$$

Then, using the inverse transformation Equation (54) of the set of values of $\{U_k(x, y, z)\}_{k=0}^n$ gives the ninth-order approximate solution as

$$
u(x, y, z, t) \approx \sum_{k=0}^{9} U_k(x, y, z)\, t^k = \sin(x + 2y + 3z)\left\{ t - \frac{t^3}{3!} + \frac{t^5}{5!} - \frac{t^7}{7!} + \frac{t^9}{9!} + \cdots \right\},
$$

which coincides with the exact solution of Equation (59) on the limit.

As an illustration for the numerical experiment using RDTM, Table 6 shows absolute and relative errors for the 3D linearized dispersive KdV equation using 5-terms of RDTM solution. Graphical representation of the exact solution and RDTM solution in 3D by setting $t = 0.5$ is shown in Figure 13.

**Table 6.** Comparison between the exact solution and five-term approximation solution via RDTM.

| $(x, y)$ | $z$ | $u_{exact}$ | $u_{RDTM}$ | Absolute Error | Relative Error |
|---|---|---|---|---|---|
| | 0.1 | 0.27070402 | 0.27070489 | $8.72219677 \times 10^{-7}$ | $3.22204181 \times 10^{-6}$ |
| (0.1, 0.1) | 0.5 | 0.46688742 | 0.46688893 | $1.50433080 \times 10^{-6}$ | $3.22204181 \times 10^{-6}$ |
| | 0.9 | 0.06765654 | 0.06765675 | $2.17992187 \times 10^{-7}$ | $3.22204181 \times 10^{-6}$ |
| | 0.1 | 0.46688742 | 0.46688893 | $1.50433080 \times 10^{-6}$ | $3.22204181 \times 10^{-6}$ |
| (0.5, 0.5) | 0.5 | 0.06765654 | 0.06765675 | $2.17992187 \times 10^{-7}$ | $3.22204181 \times 10^{-6}$ |
| | 0.9 | $-0.41785568$ | $-0.41785703$ | $1.34634848 \times 10^{-6}$ | $3.22204181 \times 10^{-6}$ |
| | 0.1 | 0.06765654 | 0.06765675 | $2.17992187 \times 10^{-7}$ | $3.22204181 \times 10^{-6}$ |
| (0.9, 0.9) | 0.5 | $-0.41785568$ | $-0.41785703$ | $1.34634848 \times 10^{-6}$ | $3.22204181 \times 10^{-6}$ |
| | 0.9 | $-0.37048303$ | $-0.37048422$ | $1.19371181 \times 10^{-6}$ | $3.22204181 \times 10^{-6}$ |

*Results at $t = 0.5$.*

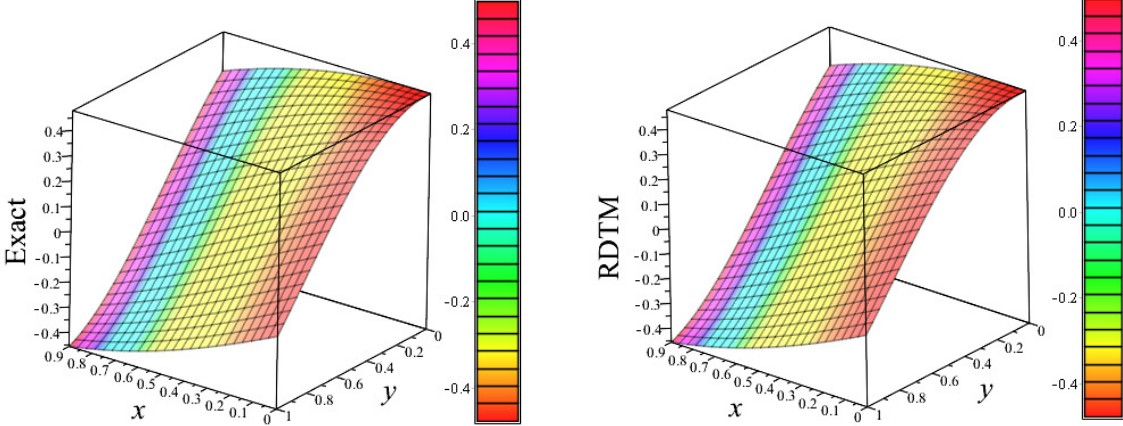

**Figure 13.** Plots of exact solution and approximate solution using RDTM vs. $x$ vs. $y$ along $z = 0.50$ and at $t = 0.50$.

## 6. Bernstein-Adomian-Laplace Decomposition Method (BALDM)

In a recent paper by Qasim and AL-Rawi [51], the authors investigated the Adomian-decomposition method based on Bernstein polynomials and with some modification. In this work, our aim here is to introduce a new modification using Laplace transform with ADM and to propose a new scheme, BALDM, which combines Bernstein's polynomial with the Adomian-Laplace decomposition method. This method can be used to solve linear and nonlinear ordinary and partial differential equations (see [51,52]). To illustrate the basic idea of BALDM, we consider

$$\mathsf{L}_t u(x, t) + R u(x, t) + N u(x, t) = g(x, t), \tag{64}$$

with initial condition

$$u(x, 0) = h(x),$$

where $\mathsf{L}_t = \frac{\partial}{\partial t}$, $R$ is a linear operator that includes partial derivatives with respect to $x$, $N$ is a nonlinear operator and $g$ is the source term. Taking the Laplace transform on both sides of Equation (64)

$$\hat{\mathcal{L}}\Big\{L_t u(x,t)\Big\} = \hat{\mathcal{L}}\Big\{g(x,t) - Ru(x,t) - Nu(x,t)\Big\}.$$

From differentiation and linearity property of Laplace transform $\hat{\mathcal{L}}$ of Equation (41) and using the decomposition series given in Equations (10) and (11), we obtain

$$\sum_{n\geq 0} \hat{\mathcal{L}}[u_n(x,t)] = \frac{h(x)}{s} + \frac{1}{s}\hat{\mathcal{L}}[g(x,t)] - \frac{1}{s}[Ru(x,t) + Nu(x,t)]. \tag{65}$$

Taking the Laplace inverse of Equation (65) yields

$$\sum_{n\geq 0} u_n(x,t) = h(x) + \hat{\mathcal{L}}^{-1}\left[\frac{1}{s}\hat{\mathcal{L}}[g(x,t)]\right] - \hat{\mathcal{L}}^{-1}\left[\frac{1}{s}\hat{\mathcal{L}}[Ru(x,t) + Nu(x,t)]\right]. \tag{66}$$

Now, by assuming that the term $g \in C[0,1]$ can be expanded in the terms of Bernstein series of order $k \in \mathbb{N}$,

$$g(t) = \sum_{i=0}^{m} \alpha_i\, B_i(t),$$

where $B_i(x)$ is the Bernstein polynomial [51]. (Please note that $C[0,1]$ denotes the space of continuous function on $[0,1]$). Comparing both sides of Equation (66), we have that

$$u_0(x,t) = h(x) + \hat{\mathcal{L}}^{-1}\left[\frac{1}{s}\hat{\mathcal{L}}\left[\sum_{i=0}^{m} \alpha_i\, B_i(t)\right]\right], \tag{67a}$$

$$u_1(x,t) = -\hat{\mathcal{L}}^{-1}\left[\frac{1}{s}\hat{\mathcal{L}}[Ru_0(x,t) + A_0(u)]\right], \tag{67b}$$

$$\vdots \tag{67c}$$

$$u_{n+1}(x,t) = -\hat{\mathcal{L}}^{-1}\left[\frac{1}{s}\hat{\mathcal{L}}[Ru_n(x,t) + A_n(u)]\right], \quad n \geq 1.$$

where $A_n$ are the Adomian polynomials first mentioned in Equation (11). We note that the approximate solution obtained via BALDM relies on the nature of the source terms and prescribed initial conditions.

### 6.1. Solution of Numerical Experiment 3

Consider the inhomogeneous linearized dispersive KdV equation

$$u_t + xu_x + u_{xxx} = 3xt^2 + 2x + xt^3, \quad x \in [0,1.0], \quad t \in [0,1.0], \tag{68}$$

with initial condition $u_0(x) = 2x$. Exact solution for Equation (68) is $u(x,t) = 2x + xt^3$, and it can be verified by using the Ansatz technique. To test the 'noise' behavior of the inhomogenous problem given in Equation (68), LADM is applied to test how the exact solution appears in the first few iterations. In the following discussion, we also introduced BALDM to solve Equation (68) to capture this self-canceling phenomena.

#### 6.1.1. Implementation of LADM for Equation (68)

By applying LADM $\hat{\mathcal{L}}$ to Equation (68), we have that

$$\hat{\mathcal{L}}[u_t] = s\,\hat{\mathcal{L}}[u(x,t)] - u(x,0) = \hat{\mathcal{L}}[3xt^2 + 2x + xt^3 - xu_x - u_{xxx}]. \tag{69}$$

Applying inverse Laplace's transform to Equation (69) gives

$$u(x,t) = \frac{1}{s} u(x,0) - \hat{\mathcal{L}}^{-1} \left[ \frac{1}{s} \left[ \mathcal{L} \left[ 3xt^2 + 2x + xt^3 - xu_x - u_{xxx} \right] \right] \right]. \tag{70}$$

Equation (70) can be given equivalently by

$$u(x,t) = u(x,0) + 6x\, \hat{\mathcal{L}}^{-1} \left[ \left( \frac{u}{s} \right)^4 \right] + 2x\, \hat{\mathcal{L}}^{-1} \left[ \left( \frac{u}{s} \right)^2 \right] + \hat{\mathcal{L}}^{-1} \left[ \frac{u}{s} \left( x \mathcal{L}[u_x] + \mathcal{L}[u_{xxx}] \right) \right]. \tag{71}$$

Substituting the decomposition series $u(x;t) = \sum\limits_{n \geq 0} u_n(x;t)$ in Equation (71) takes the form

$$\sum_{n \geq 0} u_n(x;t) = 2x + xt^3 + 2xt + \frac{xt^4}{4} + 6x\, \hat{\mathcal{L}}^{-1} \left[ \frac{u}{s} \left[ x \sum_{n \geq 0} \hat{\mathcal{L}}[u_{n,x}] + \sum_{n \geq 0} \hat{\mathcal{L}}[u_{n,xxx}] \right] \right]. \tag{72}$$

By comparing both sides of Equation (72), we obtain

$$\left. \begin{aligned} u_0(x) &= 2x + xt^3 + 2xt + \tfrac{xt^4}{4}, \\ u_1(x;t) &= \hat{\mathcal{L}}^{-1} \left[ \tfrac{1}{s} \left( x\hat{\mathcal{L}}[u_{0,x}] + \hat{\mathcal{L}}[u_{0,xxx}] \right) \right] = -2xt - \frac{xt^4}{4} - xt^2 - \frac{xt^5}{20}, \\ u_2(x;t) &= \hat{\mathcal{L}}^{-1} \left[ \tfrac{1}{s} \left( x\hat{\mathcal{L}}[u_{1,x}] + \hat{\mathcal{L}}[u_{1,xxx}] \right) \right] = xt^2 + \frac{xt^3}{3} + \frac{xt^5}{20} + \frac{xt^6}{120}, \\ u_3(x;t) &= \hat{\mathcal{L}}^{-1} \left[ \tfrac{1}{s} \left( x\hat{\mathcal{L}}[u_{2,x}] + \hat{\mathcal{L}}[u_{2,xxx}] \right) \right] = -\frac{xt^3}{3} - \frac{xt^4}{4} - \frac{xt^6}{6} - \frac{xt^7}{840}. \end{aligned} \right\} \tag{73}$$

and so on for other components. It is clear that the self-canceling 'noise' terms appear between various components and keeping the non-canceled term using Equation (10) gives immediately the exact solution

$$u(x,t) = u_0(x,t) + u_1(x,t) + u_2(x,t) + u_3(x,t) + \ldots = 2x + xt^3,$$

as this fact is already mentioned in Remark 4.

6.1.2. Solution for Equation (68) Using BALDM

To apply BALDM, first the source term $g$ is expanded in Bernstein basis of order $m = 6$ in $t$ on the interval $[0,1]$ (using Maple software package) as [51]

$$g_b(x,t) = 2.0x + 5.27778 \times 10^1 xt + 2.91667xt^2 + 5.55556 \times 10^1 xt^3.$$

By now applying BALDM, we obtain

$$u_0(x,t) = h(x) + \mathcal{L}^{-1} \left[ \frac{1}{s} \hat{\mathcal{L}}[\sum_{i=0}^{m} \alpha_i\, B_i(t)] \right],$$

$$= 2x + 1.38889 \times 10^{-1} x\, t^4 + 9.72222 \times 10^{-1} x\, t^3 + 2.63889 \times 10^{-1} x\, t^2 + 2.0\, xt,$$

$$u_1(x,t) = -\hat{\mathcal{L}}^{-1} \left[ \frac{1}{s} \hat{\mathcal{L}}\left[ x\, u_{0,x}(x,t) + u_{0,xxx}(x,t) \right] \right],$$

$$= -2.77778 \times 10^{-2} x\, t^5 - 2.43056 \times 10^{-1} x\, t^4 - 8.79630 \times 10^{-2} x\, t^3 - 1.0\, xt^2 - 2.0\, xt,$$

$$u_2(x,t) = -\hat{\mathcal{L}}^{-1} \left[ \frac{1}{s} \hat{\mathcal{L}}\left[ x\, u_{1,x}(x,t) + u_{1,xxx}(x,t) \right] \right],$$

$$= 4.62963 \times 10^{-3} x\, t^6 + 4.86111 \times 10^{-2} x\, t^5 + 2.19907 \times 10^{-2} x\, t^4 + 0.33333x\, t^3 + 1.0\, x\, t^2,$$

$$u_3(x,t) = -\hat{\mathcal{L}}^{-1}\left[\frac{1}{s}\hat{\mathcal{L}}\left[x\,u_{2,x}(x,t) + u_{2,xxx}(x,t)\right]\right],$$

$$= -6.61376 \times 10^{-4}\,x\,t^7 - 8.1019 \times 10^{-3}\,x\,t^6 - 4.39815 \times 10^{-3}\,x\,t^5 - 8.33333 \times 10^{-2}\,x\,t^4 - 0.33333\,x\,t^3,$$

$$u_4(x,t) = -\hat{\mathcal{L}}^{-1}\left[\frac{1}{s}\hat{\mathcal{L}}\left[x\,u_{3,x}(x,t) + u_{3,xxx}(x,t)\right]\right],$$

$$= 8.26720 \times 10^{-5}\,x\,t^8 + 1.15741 \times 10^{-3}\,x\,t^7 + 7.33025 \times 10^{-4}\,x\,t^6 + 1.66667 \times 10^{-2}\,x\,t^5 + 8.33333 \times 10^{-2}\,x\,t^4,$$

$$u_5(x,t) = -\hat{\mathcal{L}}^{-1}\left[\frac{1}{s}\hat{\mathcal{L}}\left[x\,u_{4,x}(x,t) + u_{4,xxx}(x,t)\right]\right],$$

$$= -9.18578 \times 10^{-6}\,x\,t^9 - 1.44676 \times 10^{-4}\,x\,t^8 - 1.04718 \times 10^{-4}\,x\,t^7 - 2.77778 \times 10^{-3}\,xt^6 - 1.66667 \times 10^{-2}\,xt^5,$$

The sum of the first five approximate solutions to the exact solution is given by

$$\Psi_5(x,t) = 2\,x - 9.18577 \times 10^{-6}\,x\,t^9 - 6.20040 \times 10^{-5}\,x\,t^8 + 3.91314 \times 10^{-4}\,x\,t^7 - 5.51698 \times 10^{-3}\,x\,t^6$$
$$+ 1.64352 \times 10^{-2}\,xt^5 - 8.21760 \times 10^{-2}\,x\,t^4 + 8.84260 \times 10^{-1}\,x\,t^3 + 2.63889 \times 10^{-1}\,x\,t^2.$$

**Remark 12.** *Table 7 compares the approximate solution, $\Psi_5(x,t) = \sum_{i=0}^{5} u_i(x,t)$, given in Equation (74) with the exact solution $u(x,t) = 2x + xt^3$. Figure 14 refers to graphical representation of the exact and 5-term approximate BALDM solution. Table 7 shows the absolute and relative errors obtained by the three-dimensional in-homogeneous dispersive KdV equation via BALDM using 5-terms are of orders of $10^{-6}$ to $10^{-2}$ for t-values in the range $t \in [0, 1.0]$ (see also Figures 15 and 16). This experiment shows BALDM can efficiently capture inhomogeneous KdV equation without exhibiting 'noise' terms, which is shown by LADM in Section 6.1.1. We also note that the numerical Tables in this paper are obtained using Equations (2) and (3).*

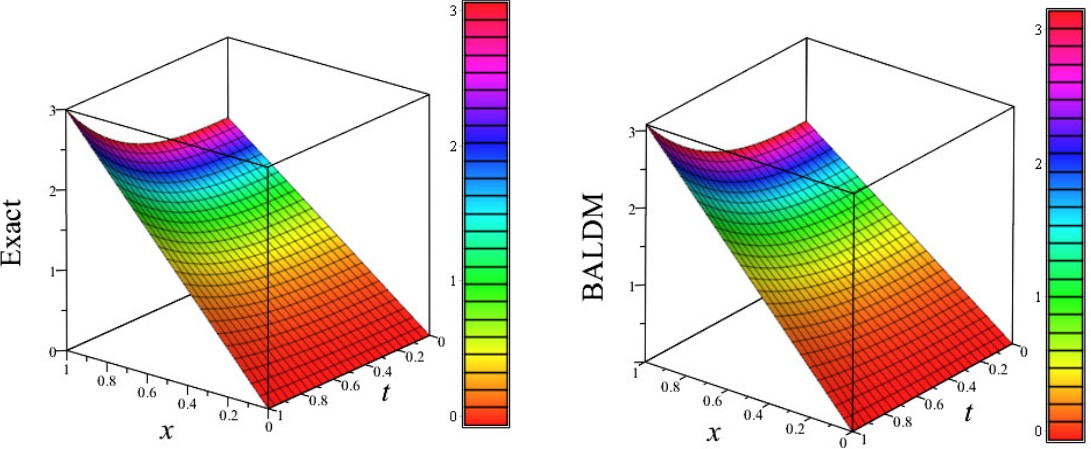

**Figure 14.** 3D plots of exact solution and sum of the first five approximate solution, $\Psi_5(x,t)$, using BALDM, vs. $x$ vs. $t$ for numerical experiment 3.

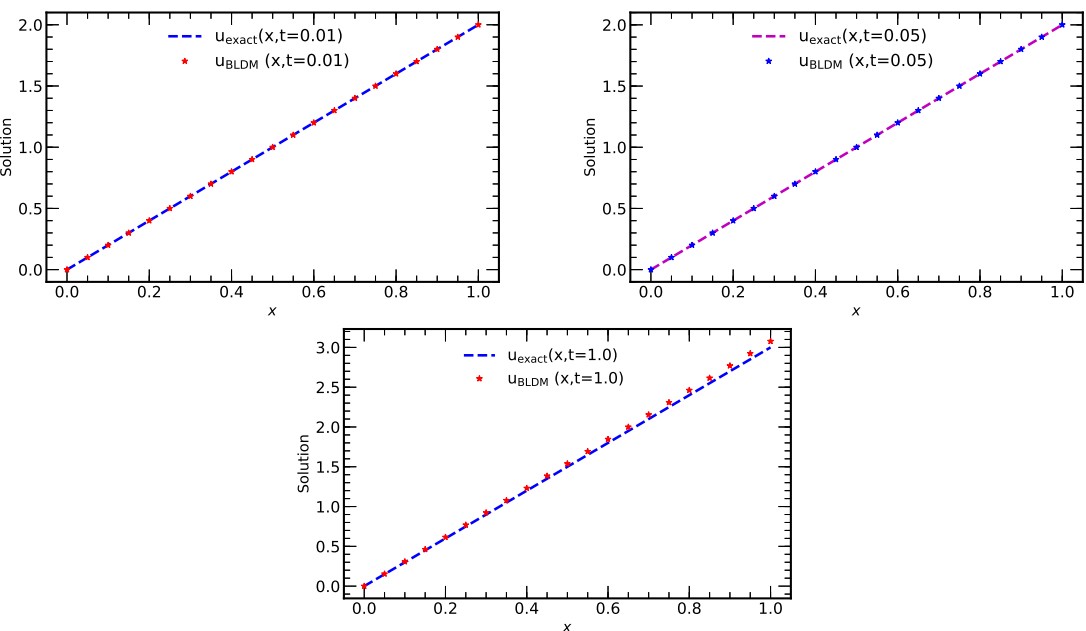

**Figure 15.** Plots of exact solution and solution using BALDM at different *t*-values, $t = 0.01, 0.05, 1.0$ (for numerical experiment 3).

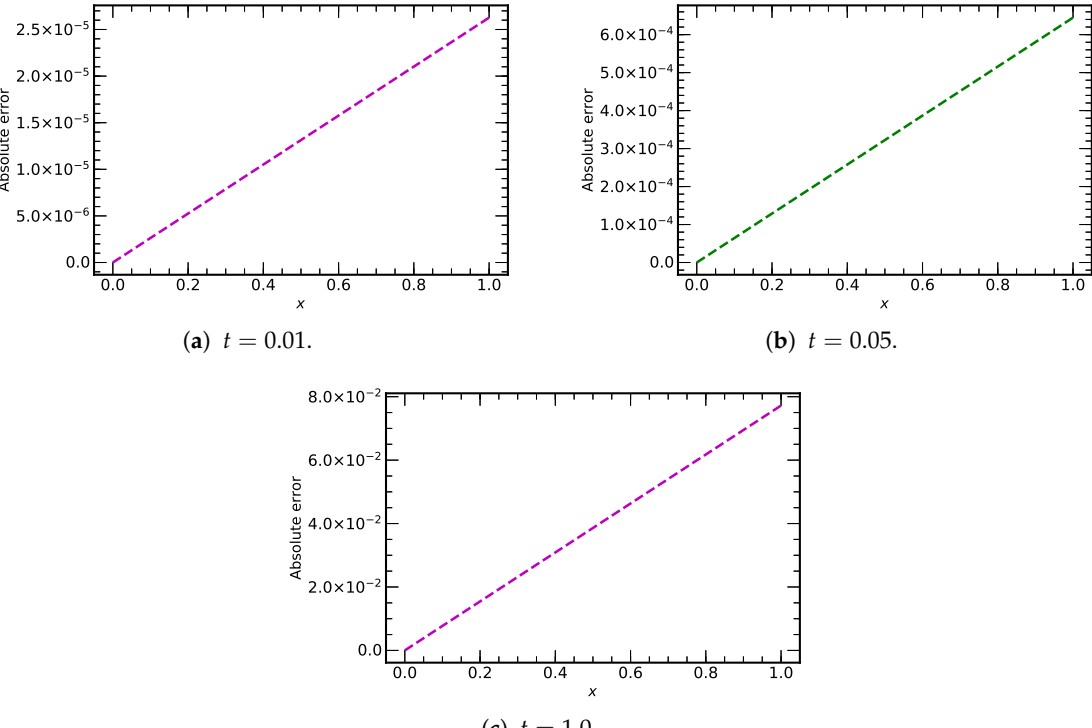

**Figure 16.** Plots of absolute errors vs. *x* using 5-terms of BALDM at different values of time ($t = 0.01, 0.05, 1.0$).

**Table 7.** Absolute and relative errors at some values of $x$ obtained at times 0.01, 0.05, 1.0 using $\Psi_5(x,t)$ of BALDM (for numerical experiment 3).

| $t$ | Values of $x$ | Exact Solution | Numerical Solution | Absolute Error | Relative Error |
|---|---|---|---|---|---|
| | 0.050 | 0.100000 | 0.100001 | $1.313616 \times 10^{-6}$ | $1.313616 \times 10^{-5}$ |
| | 0.100 | 0.200000 | 0.200003 | $2.627233 \times 10^{-6}$ | $1.313616 \times 10^{-5}$ |
| | 0.150 | 0.300000 | 0.300004 | $3.940849 \times 10^{-6}$ | $1.313616 \times 10^{-5}$ |
| | 0.200 | 0.400000 | 0.400005 | $5.254466 \times 10^{-6}$ | $1.313616 \times 10^{-5}$ |
| | 0.250 | 0.500000 | 0.500007 | $6.568082 \times 10^{-6}$ | $1.313616 \times 10^{-5}$ |
| | 0.300 | 0.600000 | 0.600008 | $7.881698 \times 10^{-6}$ | $1.313616 \times 10^{-5}$ |
| | 0.350 | 0.700000 | 0.700010 | $9.195315 \times 10^{-6}$ | $1.313616 \times 10^{-5}$ |
| | 0.400 | 0.800000 | 0.800011 | $1.050893 \times 10^{-5}$ | $1.313616 \times 10^{-5}$ |
| $t = 0.01$ | 0.450 | 0.900000 | 0.900012 | $1.182255 \times 10^{-5}$ | $1.313616 \times 10^{-5}$ |
| | 0.500 | 1.000001 | 1.000014 | $1.313616 \times 10^{-5}$ | $1.313616 \times 10^{-5}$ |
| | 0.550 | 1.100001 | 1.100015 | $1.444978 \times 10^{-5}$ | $1.313616 \times 10^{-5}$ |
| | 0.600 | 1.200001 | 1.200016 | $1.576340 \times 10^{-5}$ | $1.313616 \times 10^{-5}$ |
| | 0.650 | 1.300001 | 1.300018 | $1.707701 \times 10^{-5}$ | $1.313616 \times 10^{-5}$ |
| | 0.700 | 1.400001 | 1.400019 | $1.839063 \times 10^{-5}$ | $1.313616 \times 10^{-5}$ |
| | 0.750 | 1.500001 | 1.500020 | $1.970425 \times 10^{-5}$ | $1.313616 \times 10^{-5}$ |
| | 0.800 | 1.600001 | 1.600022 | $2.101786 \times 10^{-5}$ | $1.313616 \times 10^{-5}$ |
| | 0.850 | 1.700001 | 1.700023 | $2.233148 \times 10^{-5}$ | $1.313616 \times 10^{-5}$ |
| | 0.900 | 1.800001 | 1.800025 | $2.364510 \times 10^{-5}$ | $1.313616 \times 10^{-5}$ |
| | 0.950 | 1.900001 | 1.900026 | $2.495871 \times 10^{-5}$ | $1.313616 \times 10^{-5}$ |
| | 1.000 | 2.000001 | 2.000027 | $2.627233 \times 10^{-5}$ | $1.313616 \times 10^{-5}$ |
| | 0.050 | 0.100006 | 0.100038 | $3.223730 \times 10^{-5}$ | $3.223529 \times 10^{-4}$ |
| | 0.100 | 0.200013 | 0.200077 | $6.447461 \times 10^{-5}$ | $3.223529 \times 10^{-4}$ |
| | 0.150 | 0.300019 | 0.300115 | $9.671191 \times 10^{-5}$ | $3.223529 \times 10^{-4}$ |
| | 0.200 | 0.400025 | 0.400154 | $1.289492 \times 10^{-4}$ | $3.223529 \times 10^{-4}$ |
| | 0.250 | 0.500031 | 0.500192 | $1.611865 \times 10^{-4}$ | $3.223529 \times 10^{-4}$ |
| | 0.300 | 0.600038 | 0.600231 | $1.934238 \times 10^{-4}$ | $3.223529 \times 10^{-4}$ |
| | 0.350 | 0.700044 | 0.700269 | $2.256611 \times 10^{-4}$ | $3.223529 \times 10^{-4}$ |
| | 0.400 | 0.800050 | 0.800308 | $2.578984 \times 10^{-4}$ | $3.223529 \times 10^{-4}$ |
| | 0.450 | 0.900056 | 0.900346 | $2.901357 \times 10^{-4}$ | $3.223529 \times 10^{-4}$ |
| $t = 0.05$ | 0.500 | 1.000063 | 1.000385 | $3.223730 \times 10^{-4}$ | $3.223529 \times 10^{-4}$ |
| | 0.550 | 1.100069 | 1.100423 | $3.546103 \times 10^{-4}$ | $3.223529 \times 10^{-4}$ |
| | 0.600 | 1.200075 | 1.200462 | $3.868476 \times 10^{-4}$ | $3.223529 \times 10^{-4}$ |
| | 0.650 | 1.300081 | 1.300500 | $4.190850 \times 10^{-4}$ | $3.223529 \times 10^{-4}$ |
| | 0.700 | 1.400088 | 1.400539 | $4.513223 \times 10^{-4}$ | $3.223529 \times 10^{-4}$ |
| | 0.750 | 1.500094 | 1.500577 | $4.835596 \times 10^{-4}$ | $3.223529 \times 10^{-4}$ |
| | 0.800 | 1.600100 | 1.600616 | $5.157969 \times 10^{-4}$ | $3.223529 \times 10^{-4}$ |
| | 0.850 | 1.700106 | 1.700654 | $5.480342 \times 10^{-4}$ | $3.223529 \times 10^{-4}$ |
| | 0.900 | 1.800113 | 1.800693 | $5.802715 \times 10^{-4}$ | $3.223529 \times 10^{-4}$ |
| | 0.950 | 1.900119 | 1.900731 | $6.125088 \times 10^{-4}$ | $3.223529 \times 10^{-4}$ |
| | 1.000 | 2.000125 | 2.000770 | $6.447461 \times 10^{-4}$ | $3.223529 \times 10^{-4}$ |
| | 0.050 | 0.150000 | 0.153861 | $3.860528 \times 10^{-3}$ | $2.573685 \times 10^{-2}$ |
| | 0.100 | 0.300000 | 0.307721 | $7.721056 \times 10^{-3}$ | $2.573685 \times 10^{-2}$ |
| | 0.150 | 0.450000 | 0.461582 | $1.158158 \times 10^{-2}$ | $2.573685 \times 10^{-2}$ |
| | 0.200 | 0.600000 | 0.615442 | $1.544211 \times 10^{-2}$ | $2.573685 \times 10^{-2}$ |
| | 0.250 | 0.750000 | 0.769303 | $1.930264 \times 10^{-2}$ | $2.573685 \times 10^{-2}$ |
| | 0.300 | 0.900000 | 0.923163 | $2.316317 \times 10^{-2}$ | $2.573685 \times 10^{-2}$ |
| | 0.350 | 1.050000 | 1.077024 | $2.702369 \times 10^{-2}$ | $2.573685 \times 10^{-2}$ |
| | 0.400 | 1.200000 | 1.230884 | $3.088422 \times 10^{-2}$ | $2.573685 \times 10^{-2}$ |
| | 0.450 | 1.350000 | 1.384745 | $3.474475 \times 10^{-2}$ | $2.573685 \times 10^{-2}$ |
| $t = 1.0$ | 0.500 | 1.500000 | 1.538605 | $3.860528 \times 10^{-2}$ | $2.573685 \times 10^{-2}$ |
| | 0.550 | 1.650000 | 1.692466 | $4.246581 \times 10^{-2}$ | $2.573685 \times 10^{-2}$ |
| | 0.600 | 1.800000 | 1.846326 | $4.632633 \times 10^{-2}$ | $2.573685 \times 10^{-2}$ |
| | 0.650 | 1.950000 | 2.000187 | $5.018686 \times 10^{-2}$ | $2.573685 \times 10^{-2}$ |
| | 0.700 | 2.100000 | 2.154047 | $5.404739 \times 10^{-2}$ | $2.573685 \times 10^{-2}$ |
| | 0.750 | 2.250000 | 2.307908 | $5.790792 \times 10^{-2}$ | $2.573685 \times 10^{-2}$ |
| | 0.800 | 2.400000 | 2.461768 | $6.176845 \times 10^{-2}$ | $2.573685 \times 10^{-2}$ |
| | 0.850 | 2.550000 | 2.615629 | $6.562897 \times 10^{-2}$ | $2.573685 \times 10^{-2}$ |
| | 0.900 | 2.700000 | 2.769490 | $6.948950 \times 10^{-2}$ | $2.573685 \times 10^{-2}$ |
| | 0.950 | 2.850000 | 2.923350 | $7.335003 \times 10^{-2}$ | $2.573685 \times 10^{-2}$ |
| | 1.000 | 3.000000 | 3.077211 | $7.721056 \times 10^{-2}$ | $2.573685 \times 10^{-2}$ |

## 7. Discussion and Concluding Remarks

Motivated by their mathematical applications in numerous physical phenomena, semi-analytical solutions for dispersive KdV equations are main focus of this work as these methods are effective to provide approximations of the higher reliability of series solution. We made a comparative study of some semi-analytic methods namely; LADM with its new modification BALDM, HPM with its modification HPTM, and RDTM to solve homogeneous as well inhomogeneous linear dispersive KdV equations in 1D and higher dimensions.

In the case of homogeneous linear dispersive KdV equation, the three semi-analytic methods (LADM, HPM and RDTM) are equivalent and therefore give the same results.

In the case of inhomogeneous linear KdV equation, RDTM gives very efficient results by exhibiting rapid convergence of the series solution without showing noise terms. LADM and HPM provide the components of the series solution, where these components possess noise terms. However, the standard HPM does not produce good results, as depicted by the values of the relative errors from Table 3. Instead, we designed two modifications to alleviate this issue. First, we constructed a newly modified homotopy (50) by making a shift on the source term, which gave us better results for the inhomogeneous KdV equation given in (5). Secondly, we applied HPM combined with Laplace transform, HPTM, which also gives a reliable result for the approximate solutions of Equation (5) as shown in Section 3.4. We noticed that modification of HPM is essential for some PDEs of the likes given in Equation (5). The results obtained from these modifications demonstrate that both HPTM as well as HPM with modified homotopy provide a meaningful result for the inhomogeneous equation under consideration since the series solution via modified HPM (as well as HPTM) are in full agreement with the approximate solutions obtained via LADM. We also note that an appropriate choice of the zeroth approximation is essential for the inhomogeneous linear KdV equation in order to exhibit rapid convergence of the approximate solution to the exact solution due to the appearance of the self-canceling "noise" terms in the first few iterations as shown for both LADM and HPTM using Maple-18 software.

In the case of 3D dispersive KdV equation, the RDTM solutions are good analytical approximate solutions that converge to the exact solution rapidly. The numerical results shows that the present method are in excellent agreement with those of absolute and relative errors and the obtained solutions are shown in Table 6.

The merit of this work is in many folds, one being the derivation of a new scheme, BALDM, which is a modification of Adomian decomposition method using Bernstein polynomials and Laplace transform to solve Equation (68). The proposed technique relies on Bernstein polynomials to approximate the source term of the considered inhomogeneous PDE. Applying this modification of the source term using BALDM ensures not only an accurate solution but also it captures the self-canceling "noise" terms appearing in the case of the considered inhomogeneous equation. We also observe that the approximate solutions obtained via LADM and HPTM perform well on a certain range of $t$-values. Once these values are increased, the accuracy of the estimations becomes poor, at least for the number of terms used to approximate the solutions in the present approaches.

As a general concluding remark, from the numerical experiments of this work, we are optimistic to suggest that the semi-analytic methods used in this paper can also be applied to a more general family of KdV-type equations

$$\frac{\partial u}{\partial t} = \sum_{j=0}^{K} \alpha_j \frac{\partial^j u}{\partial x^j} + \sum_{\ell=0}^{M} \beta_\ell \frac{\partial^\ell (u^r)}{\partial x^\ell} + f(x,t), \qquad -\infty < x < \infty, \quad \text{and} \quad t > 0,$$

$$u(x,0) = g(x), \tag{74}$$

where $f$ and $g$ are given functions, $\alpha_j, \beta_\ell$ are real constants, $K, M, \ell, r \in \mathbb{Z}^+$ and $k, \ell \in \mathbb{Z}^+ \cup \{0\}$. In fact, for the homogeneous and homogeneous KdV equations considered in this paper are special cases of Equation (74) when $K = 3$, $M = 1$ and $r = 1$ and some non-negative values of $\alpha_j$ and $\beta_\ell$. Among all

these semi-analytic techniques, RDTM and BALDM are entirely comprehensible methods as they diminish the huge volume of calculations and furthermore their iteration steps towards an exact solution are direct and clear. It is good to point out that both HPM and LADM do not require small parameters in the considered equations so that the limitations of the ordinary perturbation methods can be eliminated and in this manner the computational procedures are simple and promising.

**Author Contributions:** Conceptualization, A.R.A. and A.S.K.; methodology, A.R.A. and A.S.K.; software, A.S.K.; validation, A.R.A.; formal analysis, A.R.A. and A.S.K.; investigation, A.R.A. and A.S.K.; resources, A.R.A.; data curation, A.R.A. and A.S.K.; writing-original draft preparation A.S.K. and A.R.A.; visualization, A.R.A. and A.S.K.; supervision, A.R.A.; project administration, A.R.A. and A.S.K.; funding acquisition, A.R.A. All authors have read and agreed to the published version of the manuscript.

**Funding:** A.S.K. is grateful to acknowledge to Nelson Mandela University for providing a postdoc Council Fellowship for the period March 2020 to February 2021.

**Conflicts of Interest:** The authors declare no conflict of interest.

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
