# Peer review of "On Semi-Analytical Solutions for Linearized Dispersive KdV Equations"

_mathematics, doi:10.3390/math8101769_

Round 1
Reviewer 1 Report
I attach my comments in a pdf file and also copy here:
Review
of Article: On semi-analytical solutions for linearized dispersive KdV equations
Manuscript ID: mathematics-940688
Major Comment:
I think this is an interesting paper and the authors have made a serious work which definitely deserves to be published. The main virtue of the paper is the construction of the BALDM method and also the modification of the HMP method. However, in the abstract the authors say „In this paper we construct four semi-analytic methods…” However, most methods are not constructed but only applied here. The question of which method or modification is new and which is already known and just used in the paper should be clear in the abstract (and, of course, everywhere in the paper).
Minor, but required corrections:
p.2 “The methods are derived in Sections 2 to 4 and results are displayed.” Yes, but where are they displayed? I might write something like this “The methods are derived and the results are displayed in Sections 2 to 4.”
(The little word “the” should be used more frequently, such as in p. 28: “The exact solution” and in p. 31: “are the main focus”)
p.4, Remark 1. “LADM decreases considerably huge volume of calculations.”
Does it mean that: LADM decreases the considerably huge volume of calculations which would be necessary by AMD?
p.6-7 Caption of Figure 1. and Table 1.: “using the methods LADM, HPM and RDTM.” The number of terms should be indicated, in the same way as in Table caption 2 or 7.
“Absolute and Relative errors” Relative is not with capital letter.
p.8 Remark 3. How the error at t=0 can be defined? The initial condition is given, isn’t it?
p.9 After (27): “By comparing the expressions in Eq. (27),” Comparing with what?
(28), second line, right side, numerator: I think , the -1 is missing.
p.12 Caption of Figure 6 and later: x and t should be in italic.
p.13, after (31): What is n? It might be r?
p.20, Def. 1.: The possible values of k is not clear. Are there infinite different reduced differential transform of a function? One might say “the k-th reduced differential transform”.
There are also two ‘is’ in the sentence.
- 22. top equation: I think these equations are not correct. At least the sin(x) and cos(x) terms should be there.
p.27: “We see that RDTM is straightforward having great advantage in the sense that it is less difficult to use it comparatively and it quickly converges to the exact solution”
Unfortunately the “three-dimensional inhomogeneous dispersive KdV equation” has been solved only by this RDTM method, but not by the other methods. So the basis of this comparison is not completely clear.
Fig. 13: The time is fixed for t=0.5, but at the 3D plots on of the axis is t as well. It might be y?
p.29, remark 12: “This experiment signifies that the accuracy of the series solution using BALDM increases when we increase number of terms in the approximate series solution for the PDE under consideration. This confirms that BALDM is a reliable semi-analytical method.”
This conclusion is also not straightforward as the only approximate solution has been calculated only for 5 terms.
p.32, (74): I think the letters K, k and j are confused. The k index of αk might be an error, is it j? The second summation is ? What is K?
Advices only, not required to perform:
p.28, Eq. (72b and c): It would be good to remind the reader that A are the Adomian polynomials first mentioned in (11)
Instead of such long tables, I think plots of the maximum or average errors as a function of t and/or as a function of the number of terms would be more informative.
Some parts containing equations, such as (69), are repeated. Referring only to them could reduce the length of the paper. Also (although these 3D figures are nice and impressive), when the error is small, displaying two, apparently identical figures gives not too much to the scientific value of the paper.

Author Response
Please see document enclosed.

Reviewer 2 Report
I have no major remarks to the content of the paper, the choice of methods and the content of the presented analysis. The authors efficiently use the applied tools. The paper is lavishly illustrated. The presented computational tests are clear and discussed in great detail. My only remark relates to the references. One of the cited more recent papers is by Professor Appanah R. Appadu (2017), the other by Doctor Ahmed Qasim. Over 200 papers concerning the KdV equation were published in 2019-2020, and at least half of them concern methods of solving the equation. Authors must update the bibliography.
Author Response
Please see document enclosed.

Reviewer 3 Report
Before the Editor makes a decision, I suggest that the authors must take into account the following corrections:
- A bibliographic reference is required for the original form of the equation (1).
- The authors must specify the motivation of the numerical experiment 3.
- Details on obtaining the exact solution for equation (5) are needed.
- It is not clear how the numerical data from the Table-1-Table-7 were obtained.
- The coefficients of the numerical solution of equation (5) are very bushy. It is very difficult to check their correctness!
- I think the authors need to emphasize more clearly the contribution of the manuscript from a scientific point of view.
- The results are very theoretical, very abstract. The authors must present a minimum utility of these results.
- Some editing "glitches" need to be corrected.
- I think, the author must strengthen the References section with some articles that use some similar techniques, to make the techniques used more plausible, for instance: Analytical solution of thermoelastic interaction in a half-space by pulsed laser heating, Physica E Low Dimens. Syst. Nanostruct., 87, 254-260(2017)
If the authors take into account all these corrections, then this manuscript deserves to be published.
Author Response
Please see document enclosed.
